# Nutritional supplementation, tooth crown size, and trait expression in individuals from Tezonteopan, Mexico

Erin C. Blankenship-Sefczek [1,2]*, Alan H. Goodman[3‡], Mark Hubbe[2‡], John P. Hunter[4], Debbie Guatelli-Steinberg[2]

**1** Department of Oral Biology, School of Dentistry, Creighton University, Omaha, Nebraska, United States of America, **2** Department of Anthropology, The Ohio State University, Columbus, Ohio, United States of America, **3** School of Natural Sciences, Hampshire College, Amherst, Massachusetts, United States of America, **4** Department of Evolution, Ecology and Organismal Biology, The Ohio State University, Newark, Ohio, United States of America

☉ These authors contributed equally to this work.
‡ AHG and MH also contributed to this work.
* ErinBlankenship-Sefczek@creighton.edu

**Data Availability Statement:** All relevant data are found within the manuscript and its Supporting Information files.

## Abstract

Understanding how epigenetic factors impact dental phenotypes can help refine the use of teeth for elucidating biological relationships among human populations. We explored relationships among crown size, principal cusp spacing, and accessory cusp expression in maxillary dental casts of nutritionally supplemented (n = 34) and non-supplemented (n = 39) individuals from Tezonteopan, Mexico. We hypothesized that the non-supplemented group would exhibit smaller molar crowns and reduced intercusp spacing. Since intercusp spacing is thought to be more sensitive to epigenetic influences than crown size, we predicted that the supplemented and non-supplemented groups would differ more in the former than the latter. Previous work suggests that molar accessory cusp expression may be elevated under conditions of stress. We therefore expected evidence of greater Carabelli and Cusp 5 trait expression in the non-supplemented group. We further hypothesized that anterior teeth would be affected by nutritional stress during development, with the non-supplemented group having smaller anterior tooth crowns and therefore limited space to form the tuberculum dentale. Finally, we tested whether the presence of molar accessory traits followed predictions of the Patterning Cascade Model of tooth morphogenesis in the entire sample. Our results supported the expectation that cusp spacing would differ more than molar crown size between the two groups. Carabelli trait showed little evidence of frequency differences between groups, but some evidence of greater trait scores in the non-supplemented group. The non-supplemented group also showed evidence of greater Cusp 5 frequency and expression. In the central incisors and canines, there was strong evidence for smaller crown sizes and reduced tuberculum dentale frequency in the non-supplemented group. With both groups pooled together, there was strong evidence of closer mesiodistal distances among principal cusps in molars with accessory cusps, a finding that is consistent with the PCM. Overall, our findings suggest that nutritional stress may affect accessory cusp expression.

**Funding:** The authors received no specific funding for this work.

**Competing interests:** The authors have declared no competing interests exist.

## Introduction

Characteristics of teeth are frequently used in anthropological studies to assess biological affinities among modern human populations [1–7] and phylogenetic relationships among hominin species [3, 8–17]. These studies assume that phenotypic variation in morphological features of tooth crowns is less influenced by external factors (e.g., nutritional intake and disease) than by genes. Yet, some evidence suggests that perturbations experienced during dental morphogenesis can affect aspects of dental phenotype, such as principal cusp spacing [18] and molar trait expression [19].

Dental morphology is the result of a series of events during odontogenesis, leaving open the potential for developmental perturbations to affect final tooth form. For example, nutritionally deprived pigs showed small molar sizes, decreased cusp number, and delayed development of later-forming teeth [20]. In humans, individuals with many developmental defects of enamel (specifically, linear enamel hypoplasias) expressed accessory cusps at higher frequencies and of larger sizes on first and second molars compared to individuals with no or few of these defects [19].

Although there is evidence supporting the impact of external disruptions on dental phenotypes, it remains unclear whether, and how, developmental processes are affected within this context. The Patterning Cascade Model (PCM) of dental morphogenesis can be used to explore the relationship between stress and developmental perturbation. The PCM has emerged as a powerful conceptual framework for understanding how enamel knot spacing and the space available for crown formation affect variation in dental morphology [17, 21–24]. The model predicts that because morphogenesis occurs in a developmental cascade [21], the formation of later-forming cusps will be regulated by the morphology of earlier established ones [25]. Thus, the PCM suggests that molecular interactions influencing the placement and spacing of principal cusps have a downstream effect on the formation of later-forming accessory cusps [17, 21–24].

Given the potential for external factors, such as nutritional deprivation, to impact the course of dental morphogenesis, we compared the dental morphology of two groups—one given dietary supplements, and the other not provided with these supplements—from the rural town of Tezonteopan, Mexico [26]. This comparison allows us to investigate what aspects of dental morphology are associated with dietary stress and whether associated changes in dental morphology are consistent with the PCM.

### Background: Tooth morphogenesis and environmental stress

The enamel knot, a transient cluster of densely packed, non-dividing epithelial cells at the center of the tooth germ [27], begins to form in the late bud stage of odontogenesis [28]. The primary enamel knot forms first, followed by secondary enamel knots [28]. Molecules originating in the mesenchyme create a feedback loop with the enamel knot to signal cell proliferation activation and inhibition. Later-forming secondary enamel knots can only form at a distance from existing enamel knots, where the intensity of the molecular inhibitory field is attenuated [29–31].

The PCM proposes that molar cusp formation is regulated by interactions between the timing of enamel knot formation, the size of inhibition fields around enamel knots, and the duration of crown development [21, 25, 32]. An understanding of variation in later-forming cusps requires an analysis of the interplay between crown size, as a proxy for the duration or rate of growth and the opportunity window for new cusps to form, and intercusp distances, as an indicator of the size of inhibitory zones during development.

Studies have tested and supported different aspects of the PCM's predictions for the dentitions of rodents [25, 33, 34] and seals [21], where timing and orientation of secondary enamel knots were used to predict placement of cusps and molar morphology. For application to quadritubercular molars, various principal cusp configurations have been proposed to increase the likelihood of forming accessory cusps [17, 22, 23]. Smaller intercusp distances relative to tooth size were found to be associated with an increase in accessory cusps in general [24]. Ortiz et al. [17] categorized accessory cusps based on their locations, as either peripheral (e.g., Carabelli trait) or central (e.g., cusp 5 trait). Peripheral cusps are on the crown margin, and central cusps are located between principal cusps. Ortiz et al. [17] associated particular configurations with a greater likelihood of forming peripheral or central accessory cusps or both. Carabelli trait, found on the lingual margin of the protocone, can be considered a peripheral cusp. Cusp 5, located between the metacone and hypocone, is also positioned near the distal occlusal margin. Therefore, cusp 5 could be considered either a central or a peripheral accessory cusp and could be associated with principal cusp configurations otherwise attributed to peripheral cusps. To consider this possibility, Fig 1 illustrates configurations of secondary enamel knots (specifically principal cusps) thought to promote the formation of accessory cusps that have been tested in previous studies (Configurations A-C) [17, 22, 23] as well as a fourth configuration (Configuration D).

For non-human primates and fossil hominins [17, 23], as well as modern humans [16, 22, 23, 35], some PCM-derived expectations have received mixed support. There are various potential reasons why PCM predications have not always been met. It may be that not all configurations that promote the initiation of accessory cusps have been identified and considered. Another possibility could be sample size limitations. Small differences in cusp spacing may be difficult to detect statistically in small samples but can still be relevant to the formation of accessory cusps. In addition, as Riga et al. [19] suggested, environmental perturbations during morphogenesis might introduce trait variability that is not predicted based on the PCM alone. The present study further explores this last possibility.

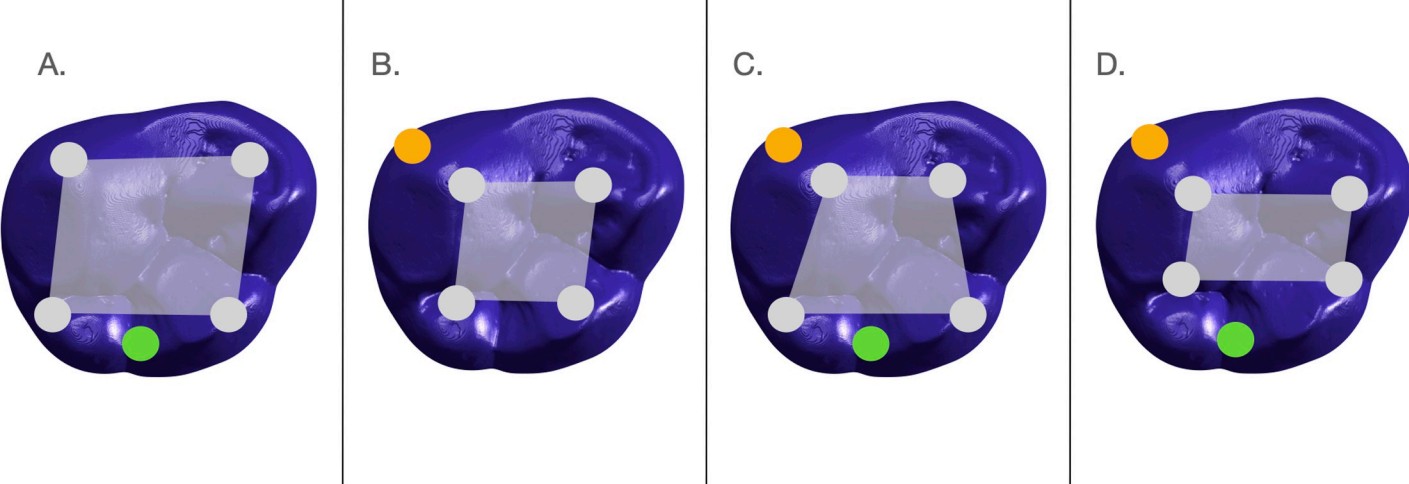

**Fig 1. Visual summary of four configurations of secondary enamel knots associated with principal cusp placement favoring the presence of Carabelli and/or cusp 5 accessory traits.** Configuration A: All principal cusps are relatively distant from each other, favoring formation of cusp 5 trait. Configuration B: All principal cusps are relatively closely spaced, favoring the formation of Carabelli trait. Configuration C: Mesial principal cusps are relatively close and distal principal cusps are relatively far, favoring the formation of both Carabelli and cusp 5 trait formation. Configuration D: Buccal and lingual principal cusps are relatively closely spaced, favoring the formation of Carabelli and/or cusp 5 trait. Configurations A-C are from Ortiz et al [17], and configuration D is a new possibility that we propose.

As found in experimental studies on nonhuman animals [36–39], there is evidence in humans that nutritional stress can disrupt morphogenesis and thus alter crown size and morphology. With respect to crown size, individuals with low-birth weight, indicative of inadequate gestational conditions, were found to exhibit reduced permanent tooth size [40]. With respect to crown morphology, the study of Riga et al. [19], assessing the association of linear enamel hypoplasia (LEH; a dental indicator of physiological stress during development) and maxillary molar cusp expression, found that individuals in the "stressed" group (those with three or more LEH defects) exhibited greater variation in molar cusp expression. Finally, though not assessing nutritional stress per se, Townsend and colleagues [18] found greater fluctuating asymmetry in average distances among principal cusps than for crown size, suggesting that the placement of primary and secondary enamel knots are more affected by epigenetic influences than is crown size.

The present study uses a unique sample to assess how crown size, principal cusp spacing, and accessory cusp frequency and expression are affected under conditions of stress. To do so, we test for differences between supplemented and non-supplemented groups in these dental parameters. The present study also evaluates whether predictions derived from PCM principles can account for differences in dental trait expression between the supplemented and non-supplemented groups. Finally, this study tests whether the presence of accessory cusps in the entire sample is associated with intercusp spacing differences expected under the PCM.

## Hypotheses

### Dietary stress, molar crown size, and spacing among principal cusps

Hypothesis 1: Dietary stress impedes growth of the molars such that molars grow more slowly or for a shorter period of time, resulting in smaller molars overall. Hypothesis 1 predicts that relative to supplemented individuals, non-supplemented individuals, which were "mildly to moderately malnourished" [26] will have smaller molar crown sizes.

Hypothesis 2: Dietary stress interferes with the process of molar morphogenesis by altering the placement of the main molar cusps. In our study sample we expected that the non-supplemented group would have smaller absolute intercusp distances as well as smaller crown sizes (as per Hypothesis 1). Based on Townsend et al. [18], intercusp distances are expected to be more affected by epigenetic factors than crown size. Hypothesis 2 predicts that the non-supplemented group will have both smaller absolute and relative principal intercusp distances compared to the supplemented group.

### Dietary stress and molar accessory cusp expression

Hypothesis 3: Dietary stress interferes with molar morphogenesis in such a way as to increase the probability of forming Carabelli trait. Hypothesis 3 predicts that the non-supplemented group will exhibit Carabelli trait at a higher frequency (Prediction 3a) and a tendency toward greater ASUDAS scores (Prediction 3b). This hypothesis and its corresponding predictions are based on the positive association between environmental stress and accessory cusp expression reported by Riga et al. [19]. Furthermore, if consistent with PCM expectations as we have laid them out in Fig 1, we would expect to see greater frequency and expression of the Carabelli trait to be associated with a configuration that is thought to promote its formation, such as Configurations B, C and/or D in Fig 1.

Hypothesis 4: For the same reasons as those given in the previous hypothesis, Hypothesis 4 predicts that relative to the supplemented group the non-supplemented group will exhibit cusp 5 at a higher frequency (Prediction 4a) and a tendency toward larger ASUDAS scores (Prediction 4b). If consistent with PCM expectations (Fig 1), this greater frequency and

expression of cusp 5 will be accompanied by cusp configurations that would promote its formation, such as Fig 1 Configurations A, C and/or D.

### Dietary stress and the anterior dentition

Hypothesis 5: Dietary stress interferes with the growth of anterior teeth resulting in smaller incisors and canines. Hypothesis 5 predicts that anterior teeth will be smaller in the non-supplemented group relative to the supplemented group, if they have a response to nutritional stress similar to that of molars (Prediction 5a). If secondary knots can form in the development of anterior teeth in humans, then larger teeth might offer a greater opportunity for secondary enamel knots to form low on the crown. In light of this possibility, we predict that morphological features such as tuberculum dentale will occur at lower frequency in the non-supplemented group where we predict crown size will be smaller (Prediction 5b).

### Molar trait expression and the Patterning Cascade Model in the entire sample

Hypothesis 6: Molar trait expression in the entire sample will be consistent with the PCM as visualized in Fig 1. All relative and absolute principal cusp spacings are tested for differences between molars with and without the respective traits. For the Carabelli trait, we predict principal cusp spacing will be consistent with Fig 1 Configurations B, C and/or D. For the cusp 5 trait, we predict principal cusp spacing will be consistent with Fig 1 Configurations A, C and/ or D. In only a small percentage of our sample were both traits present on the same tooth, limiting our ability to detect configurations associated with Carabelli and cusp 5 co-occurrence.

## Materials and methods

### Study collection

In 1968, the Mexican National Institute of Nutrition offered a program called Proyecto Puebla that gave food supplements to some families in rural communities [26]. To address the impact of nutritional intake on growth and development, Dr. Adolfo Chavez and nutritionist Celia Martinez began a longitudinal study in the poor, rural community of Tezonteopan, Mexico in 1969. Over an eleven-year period (1969–1980), about 80 subadults were monitored from infancy through adolescence for dietary habits, and growth parameters such as height, weight, and body mass index [26].

Chavez and Martinez chose the town of Tezonteopan for their study because residents shared similar socioeconomic situations and traditional dietary habits (primary consumption of maize and beans, with limited amounts of fresh vegetables) and as a community, were documented as mild-to-moderately malnourished [26]. Monitoring mothers' diets and the subsequent growth of their children was the focus for two reasons: 1) the maternal environment is known to greatly impact fetal growth, and 2) women in this community relied on breastmilk as a primary food source for the first two years of a child's life [26]. Women in the supplemented group began receiving dietary supplementation consisting of milk fortified with vitamins and minerals after the first missed menstrual cycle and continuing through the duration of their pregnancies and periods of breastfeeding [26]. Women in the non-supplemented group consumed the traditional diet of the community without additional nutritional aid [26]. Subadult participants were placed in one of two study groups: 1) a "supplemented" group consisting of more than 40 individuals who received dietary supplementation and whose mothers were given nutritional aid during pregnancy and lactation, and 2) a "non-supplemented"

group consisting of more than 40 individuals and their mothers who received no added dietary components [26].

Throughout the study, Chavez and Martinez [26] found that members of both the non-supplemented and supplemented groups experienced bouts of malnutrition, likely caused by recurrent infections and a poor base diet. Comparisons of the two groups showed that sub-adults in the non-supplemented group had lower birth weights, shorter stature, and delayed bone maturation compared to members of the supplemented group [26]. Clear distinctions in growth rate and weight gain appeared between weeks 16 and 24 after birth, with individuals in the supplemented group showing greater gains in both categories compared to members of the non-supplemented group [26]. In addition, a study by Goodman and colleagues [41] found elevated levels of LEH in the non-supplemented group suggesting this group experienced greater stress compared to individuals in the supplemented group.

As part of their study, Goodman et al. [41] took dental impressions from the participants to document the presence of LEH. Data for the present study were collected from these casts of the original study participants, not directly from living human subjects. An IRB application was submitted through Buck-IRB at Ohio State University's Office of Responsible Research Practices in 2015. Given the nature of the observation materials (casts and not living subjects), Buck-IRB stated that IRB approval was not required for study.

Casts with at least one permanent molar were used. All dentitions were those of subadults and young adults ranging in age from 10–20 years, so there was very little wear overall. From this sample, only teeth with less than 5% estimated occlusal wear were included in the analysis. With this criterion, there were 34 individuals in the supplemented group and 39 in the non-supplemented group, for a total of 73 individuals included in the analyses. Given the age of participants, very few deciduous teeth were present. Additionally, permanent teeth, particularly M1s, have been identified as key tooth types for observing accessory traits [42]. For these reasons, only permanent maxillary dentitions were included in the present study: characteristics of upper central incisors (I1), lateral incisors (I2), canines (C), first molars (M1) and second molars (M2) were recorded.

Observations were undertaken by the first author, who was blinded to the groups during data recording. Because the subjects in this study were subadults and young adults with mixed dentitions, not every tooth type mentioned above was present for each cast. Only first and second upper molars were included for observation because third molars had not yet fully erupted.

## Tooth crown size

Maximum buccolingual (BL) and mesiodistal (MD) lengths were recorded for maxillary dentitions using digital calipers [43]. Each measurement was taken three times per tooth and the means for each length were used for statistical analyses. BL and MD measurements were then multiplied (BL × MD) to estimate crown areas, and the square root of the area was taken in order to create a unitless ratio for comparison to cusp spacing (described below) and crown size [see 22].

## Molar pricipal cusp distances

As described above, cusp tip distances can be used to approximate the spacing of enamel knots during morphogenesis. Here, distances between pairs of principal cusp tips were measured using digital calipers. Six distances were measured on each tooth (Fig 2). Each distance was measured three times, and the average for each was used for statistical analyses. Relative intercusp distances (correcting for crown size), were calculated using the absolute intercusp

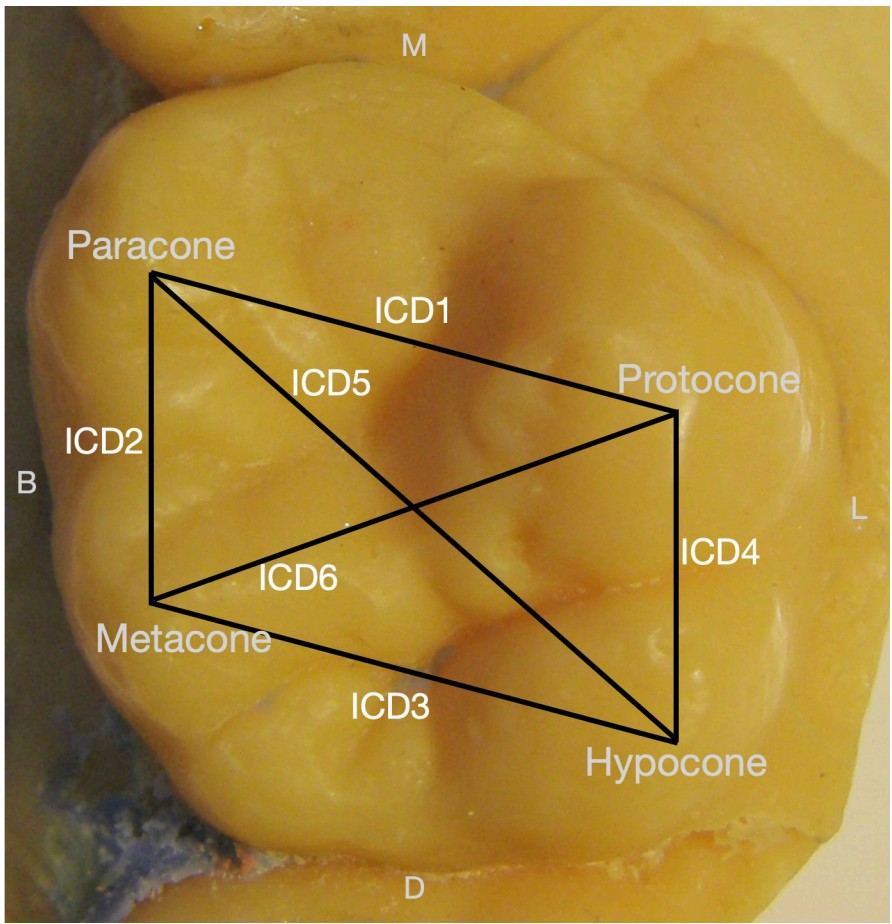

**Fig 2. Distances between upper molar principal cusps measured for comparison (ICD1-ICD6).** ICDs represent dimensions between paracone-protocone (ICD1), paracone-metacone (ICD2), metacone-hypocone (ICD3), protocone-hypocone (ICD4), paracone-hypocone (ICD5), and protocone-metacone (ICD6). Note: Lines indicate the distance measured between principal cusps and are labeled as intercusp distance (ICD) 1–6. M = mesial, D = distal, B = buccal, L = lingual.

distance divided by the square root of the corresponding molar area (BL × MD) [following 16,17]. Both measurements were used for statistical analyses.

## Tooth traits

The ASUDAS was used to score maxillary dental traits with reference to scoring plaques [44]. Recording was based on a scale of expression, beginning with "0" as absent and increasing as trait expression becomes larger. Incisors, canines, first molars, and second molars were recorded for traits and included in statistical analyses. For anterior teeth, tuberculum dentale (0–6) was recorded (Fig 3). For molar teeth cusp 5 (0–5) and Carabelli trait (0–7) were included (Fig 3). The ranges of recorded sizes of these traits allows researchers to go beyond presence/absence and allows for greater reliability in observations [44]. For some statistical analyses, trait frequency (presence/absence) was compared. The breakpoint for presence of tuberculum dentale were set at $\geq 1$. In relation to global averages, Indigenous Central and South American populations show lower frequencies of molar accessory cusps [45]. Therefore, breakpoints for Carabelli trait and cusp 5 expression in this sample were set at $\geq 1$ [42]. As the

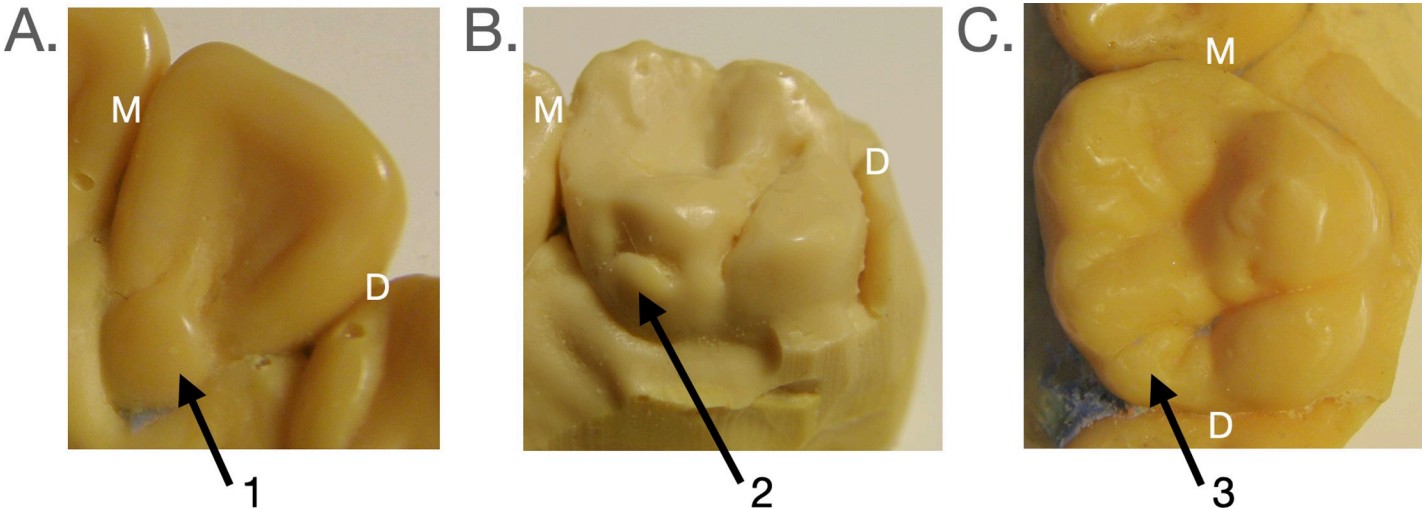

**Fig 3. Anterior (incisor and canine) and posterior (molar) tooth traits scored.** Image A: lateral incisor with tuberculum dentale (arrow #1). Image B: first molar with Carabelli trait (arrow #2). Image C: first molar with cusp 5 trait (arrow #3). For each image, M = mesial, D = distal.

first author recorded all data presented here, the One-way Interclass Correlation (ICC) statistic was calculated on a sample of 30 individuals to test intraobserver reliability overall for all traits scored in this study. Using the scale set by Koo and Li [46], results of the ICC (0.87) suggest intraobserver reliability is "good" (between 0.75 and 0.90).

## Statistical tests

All statistical tests, except for the tests for Carabelli and cusp 5 trait expression, were conducted with version 3.4.1 of R. A Shapiro-Wilks test for normality showed the data were not normally distributed (S1 Table). Therefore, nonparametric statistical tests were used. We compared supplemented and non-supplemented groups in hypotheses 1–5, and we compared morphological groups for the entire pooled sample in hypothesis 6. Given that all the hypotheses are expressed in directional terms, one-tailed tests were used throughout. Two-sample Wilcoxon tests were used to analyze differences in crown size, principal cusp spacing, and trait frequency (presence/absence). Cliff's Delta statistic was used to compute effect size and confidence interval for Carabelli trait, cusp 5, and tuberculum dentale frequency. For both the Carabelli trait and cusp 5, additional tests were conducted in SAS 9.4 to evaluate whether there were differences in trait expression between the supplemented and non-supplemented groups. For these tests, ordinal logistic regression was used to analyze how the samples differed in terms of their trait scores. In other words, these were tests of whether there were differences between the supplemented and non-supplemented groups for ASUDAS score categories of Carabelli and cusp 5 expression.

The variables analyzed for this study (crown size, intercusp spacing, and trait expression) are associated with tightly controlled developmental processes constrained by the demands of precise occlusion. Although these variables would be expected to show developmental integration, a significant portion of their expression could result from independent factors, leading to small covariances in their expression, as is frequently observed with other small biological traits [47]. Thus, statistically analyzing each trait individually helps to quantify underlying semi-independent processes that contribute to alterations in final tooth form.

Hunter and colleagues [22] found generally low-to-moderate levels of association between Carabelli ASU score and relative intercusp distance (Kendall's tao ≈ -0.3), suggesting small-to-

medium effect size of cusp spacing on Carabelli expression in a much larger sample of 187 individuals. Hunter and colleagues [22] also reported an odds ratio of approximately 8:1 from a logistic regression of Carabelli ASU score vs. relative cusp spacing, which suggests a relatively large effect when calculated over a range of 0.1 units of the relative cusp spacing ratio, which is about half of the observed range of 0.2 relative intercusp distance units between a minimum of 0.5 and a maximum of 0.7. In contrast, differences in the cusp spacing ratio observed in this study between supplemented and non-supplemented groups (see relative intercusp spacing values in Tables 2, 8 and 9 below) are much smaller than 0.2 or even 0.1, and instead range from a minimum of 0.005 to a maximum of 0.04, with most differences between groups below 0.02. Calculated over a 0.02 difference in cusp spacing, the odds ratio drops to 1.5:1, which can be considered a weak effect. Thus, we expect generally small effects on Carabelli expression and presumably other accessory features in this study. With small to medium effect sizes, larger *P*-values that diverge from the small values traditionally associated with "statistical significance", are also expected.

Given the expected effect sizes and *P*-values, to discern patterns in our results, we adopt the approach outlined by Muff and colleagues [48] where *P*-values are used to suggest the relative strength of evidence of predicted relationships rather than as binary cut-offs. Based on the Muff et al. [48] schema, *P*-values greater than 0.1 imply "little or no evidence", between 0.1 and 0.05 "weak evidence", between 0.05 and 0.01 "moderate evidence", between 0.01 and 0.001 "strong evidence", and less than 0.001 "very strong evidence". As stated by Muff and colleagues [49], "little to no evidence" does not mean an absence of evidence or that no effect exists. Due to the rarity of this sample, in which comparisons between nutritional stress levels in humans can be made, our aim is to detect possible trends. We recognize that there exists a continuing debate among statisticians regarding the statistical measure of *P*-values, including the use of *P*-values as "strength of evidence", and the application of strict α-levels [see 50, 51, 52, 53, 54, 55]. As suggested by Muff and colleagues [48, 49], we present *P*-values in conjunction with effect size to show patterns in our results and interpret the possible relationships. We recognize that our data set arises from a small sample size and interpret our findings within this framework.

## Results

### Hypothesis 1: Dietary stress and molar crown size

Hypothesis 1 predicts that the non-supplemented group will have smaller molar crowns compared to the supplemented group. Results are shown in Table 1. We found little evidence (p≥ 0.1) that M1 is smaller in the non-supplemented group, and only weak evidence (0.1>p>0.05)

**Table 1. Wilcoxon test results showing first molar (M1) and second molar (M2) crown size differences between the supplemented and non-supplemented groups.**

| | W | df | p-value | Supplemented Mean Area | Non-Supplemented Mean Area | Differences in mean areas (supplemented–non-supplemented) | 95% Confidence Interval | Effect Size* |
|---|---|---|---|---|---|---|---|---|
| | | | | | Comparing Groups | | | |
| M1 | 1896.6 | 1 | 0.176 | 116.73 | 115.30 | 1.43 | -0.289; 0.292 | 0.00879 (very small) |
| M2 | 71 | 1 | 0.064 | 110.05 | 99.74 | 10.31 | -0.400; 1.823 | 0.113 (small) |

*Effect size magnitude based on Cohen [56] where |r| <0.1: very small effect; |r| = 0.1: small effect; |r| = 0.3: medium effect, and |r| = 0.5: large effect

that M2 is smaller in the non-supplemented group. Therefore, there is little evidence in support of Hypothesis 1 from our data.

## Hypothesis 2: Dietary stress and spacing between molar cusps

Hypothesis 2 predicts that there will be smaller absolute and relative principal cusp spacing in the non-supplemented group compared to the supplemented group. Results for intercusp distances are presented in Table 2. We found that, compared to the supplemented group, the non-supplemented group showed little to no evidence ($p \geq 0.1$) of smaller absolute or relative ICDs1-4, weak evidence ($0.1 > p > 0.05$) of smaller ICD5, and moderate evidence ($0.05 > p > 0.01$) of smaller ICD6. We noted that all of the differences in absolute cusp spacing and five of the six relative cusp spacings, though small in magnitude, were positive, with the non-supplemented group being smaller. We therefore performed a sign test to explore the consistency of a possible directional effect. Considering absolute cusp spacing, the probability of obtaining one negative and six positive differences (supplemented–non-supplemented) when the predicted differences would be three negatives and three positives (i.e., true proportion 0.5) is 0.015625, which we consider to be moderate to strong evidence for a directional effect overall. Similarly, for relative

**Table 2. Wilcoxon test results showing differences in absolute cusp distances (above) and relative cusp distances (below) between the supplemented and non-supplemented groups.**

| | | | | | | Comparing Groups | | | |
|---|---|---|---|---|---|---|---|---|---|
| | Intercusp Distance Line (ICD[1]) | W | df | *p*-value | Supplemented Mean ICD | Non-Supplemented Mean ICD | Differences in mean distances (supplemented–non-supplemented) | 95% Confidence Interval | Effect Size* |
| Absolute Cusp Distances | ICD1 | 1157 | 1 | 0.296 | 7.771 | 7.617 | 0.154 | -0.094; 0.348 | 0.113 (small) |
| | ICD2 | 1197.5 | 1 | 0.173 | 5.605 | 5.447 | 0.158 | -0.134; 0.390 | 0.113 (small) |
| | ICD3 | 1078 | 1 | 0.192 | 6.926 | 6.725 | 0.201 | -7.095; 5.149 | 0.212 (small) |
| | ICD4 | 1162.45 | 1 | 0.277 | 5.216 | 5.198 | 0.018 | 0.008–0.423 | 0.219 (small) |
| | ICD5 | 1133 | 1 | 0.076 | 9.767 | 9.503 | 0.264 | -0.017; 0.531 | 0.198 (small) |
| | ICD6 | 1155.5 | 1 | 0.042 | 8.288 | 8.027 | 0.261 | 0.039; 0.529 | 0.241 (small) |
| Relative Cusp Distances | ICD1 | 1062 | 1 | 0.663 | 0.722 | 0.717 | 0.005 | -0.015; 0.029 | 0.0860 (very small) |
| | ICD2 | 1082 | 1 | 0.552 | 0.522 | 0.513 | 0.009 | -0.017; 0.031 | 0.0606 (very small) |
| | ICD3 | 1036 | 1 | 0.269 | 0.644 | 0.629 | 0.015 | -0.002; 0.041 | 0.186 (small) |
| | ICD4 | 1094 | 1 | 0.490 | 0.485 | 0.492 | -0.007 | -0.002; 0.032 | 0.181 (small) |
| | ICD5 | 1113 | 1 | 0.077 | 0.909 | 0.888 | 0.021 | -0.003; 0.040 | 0.176 (small) |
| | ICD6 | 1141 | 1 | 0.043 | 0.771 | 0.752 | 0.019 | 0.003; 0.042 | 0.246 (small) |

[1]"ICD" = Intercusp Distance. Refer to Fig 2 for distances.

*Effect size magnitude based on Cohen [56] where |r| <0.1: very small effect; |r| = 0.1: small effect; |r| = 0.3: medium effect, and |r| = 0.5: large effect

cusp spacing, the probability of obtaining one negative and five positive differences is 0.03125, which we consider to be moderate evidence for a directional effect. Findings from the sign test suggest the non-supplemented group has slightly smaller cusp spacing, in all but one (relative ICD4) dimension compared to the supplemented group.

## Hypothesis 3: Dietary stress, Carabelli trait frequency (Prediction 3a) and expression (Prediction 3b)

Hypothesis 3 predicts that compared to the supplemented group, the non-supplemented group will exhibit higher frequency (Prediction 3a) and greater expression (Prediction 3b) of Carabelli trait. Results from the comparison of molar trait frequencies are presented in Table 3. We found that, compared to the supplemented group, the non-supplemented group showed little to no evidence (p≥0.1) of a difference in Carabelli trait frequency. We noted that, based on the distribution of ASUDAS scores for first molars (S2 Table), the non-supplemented group appeared to exhibit greater degrees of trait expression compared to the supplemented group. We explored this apparent difference further, using ordinal logistic regression, in which ASUDAS scores for Carabelli trait were compared between supplemented and non-supplemented groups (Table 4). ASUDAS score was modelled as a function of supplementation vs. non-supplementation using a non-parallel slopes correction. We found moderate evidence ($p = 0.0335$; $0.05 > p > 0.01$) supporting the overall effect of nutritional supplementation on Carabelli trait expression. More specifically, we found that compared to the supplemented group, the non-supplemented group showed strong evidence ($0.01 > p > 0.001$) for elevated trait expression for the transition of ASUDAS scores 1 to 2, and weak evidence ($0.1 > p > 0.05$) for elevated trait expression for the transitions between scores 2 and 3 and between scores 4 and 5. The Odds Ratio showed the non-supplemented group had three times greater odds of having a Carabelli trait score of 2 vs a score of less than 2 compared to the supplemented group. This was the only transition in which the 95% confidence limits around the odds ratio did not include 1.

## Hypothesis 4: Dietary stress and upper cusp 5 frequency (Prediction 4a) and expression (Prediction 4b)

Hypothesis 4 predicts that relative to the supplemented group, the non-supplemented group will exhibit higher frequency (Prediction 4a) and greater expression (Prediction 4b) of upper molar cusp 5. Results of molar trait frequency are presented in Table 3. Compared to the

**Table 3. Wilcoxon test results showing differences in cusp frequency of first molars (M1) and second molars (M2) between the supplemented and non-supplemented groups and Cliff's Delta effect size.**

| | | Comparing groups | | | | |
|---|---|---|---|---|---|---|
| Tooth Type | Trait | W | Df | *p*-value | 95% Confidence Interval | *delta** |
| M1 | Carabelli | 1992.5 | 1 | 0.501 | 0.0161–0.6562 | 0.3809 (medium) |
| | Cusp 5 | 1864.5 | 1 | 0.028 | -0.5787–0.2423 | -0.4251 (medium) |
| M2 | Carabelli | 17 | 1 | 0.518 | -0.7148–0.9079 | 0.3000 (small |

**delta* magnitude based on thresholds outlined by Romano et al. [57] where |*delta*| <0.146 is a negligible effect; |*delta*| <0.33 is a small effect; |*delta*| < 0.474 is a medium effect, and >0.474 is a large effect

Note: statistical results presented here are based on data shown in S2

**Table 4. Ordinal logistic regression analysis of maximum likelihood estimates for Carabelli trait score transitions for supplemented compared to non-supplemented groups.**

| Score Transition | df | Estimate | Standard Error | Wald Chi-square | $p$ > Chi-square | 95% Confidence Limits | Odds Ratio Point Estimate* |
|---|---|---|---|---|---|---|---|
| 4→5 | 1 | 0.4984 | 0.3661 | 1.8531 | 0.0867 | 0.645–11.382 | 2.710 (medium) |
| 3→4 | 1 | 0.2750 | 0.2556 | 1.1576 | 0.1410 | 0.636–4.721 | 1.733 (small) |
| 2→3 | 1 | 0.3466 | 0.2349 | 2.1775 | 0.0700 | 0.797–5.022 | 2.000 (medium) |
| 1→2 | 1 | 0.5620 | 0.2373 | 5.6079 | 0.0089 | 1.214–7.800 | 3.077 (large) |
| 0→1 | 1 | 0.0435 | 0.2492 | 0.0305 | 0.4307 | 0.411–2.897 | 1.091 (small) |

*Effect size magnitude based on recommendations by Sullivan and Feinn [58] where OR of 1.5 is a small effect; OR of 2 is a medium effect and OR of 3 is a large effect. The sample size for the 5→6 transition was too small and could not be calculated.

supplemented group, the non-supplemented group showed moderate evidence (0.05>p>0.01) for a greater frequency of cusp 5 trait frequency in first molars. There were no cusp 5s present on second molars (M2s) in either group.

We used ordinal logistic regression to explore differences in trait expression between the two groups (Table 5). When an ordinal logistic regression was performed on the raw ASUDAS scores for cusp 5, the model did not converge, likely because of small sample sizes per cell. Thus, cusp 5 scores were concatenated into the following categories: Category 1 (raw score of 0 or absent of cusp 5), Category 2 (raw scores of 1–3 or "slight expression" of cusp 5) and Category 3 (raw scores of 4–5 or "cuspal" forms of expression). For this analysis, we did not need to use the non-parallel slopes correction, as slopes for each transition were equivalent. We found strong evidence supporting the overall effect of nutritional supplementation on cusp 5 expression ($p$ = 0.00875; 0.01>p>0.001). More specifically, compared to the supplemented group, the non-supplemented group showed strong evidence (0.01>p>0.001) for elevated cusp 5 trait expression in the transition between absent and slight, and weak to moderate (0.1>p>0.01) evidence for elevated trait expression between slight and cuspal. Overall, the Odds Ratio showed that the non-supplemented group had four times greater odds of having elevated cusp 5 expression compared to the supplemented group.

## Hypothesis 5: Dietary stress and anterior tooth crown size (Prediction 5a) and trait expression (Prediction 5b)

Hypothesis 5 predicts that anterior teeth in the non-supplemented group will be smaller (Prediction 5a) and possess the basal accessory trait tuberculum dentale at a lower frequency

**Table 5. Ordinal logistic regression analysis of maximum likelihood estimates for cusp 5 category transitions for supplemented compared to non-supplemented groups.**

| Score Transition | df | Estimate | Standard Error | Wald Chi-square | $p$ > Chi-square | Overall** 95% Confidence Limits | Overall** Odds Ratio Point Estimate* |
|---|---|---|---|---|---|---|---|
| Slight to cuspal | 1 | 0.8959 | 0.5535 | 2.6197 | 0.0527 | 1.538–10.704 | 4.058 (large) |
| Absent to slight | 1 | 0.6816 | 0.2509 | 7.3774 | 0.0033 | | |

*Effect size magnitude based on recommendations by Sullivan and Feinn [58] where OR of 1.5 is a small effect; OR of 2 is a medium effect and OR of 3 is a large effect.
** One overall value is reported since the slope is the same for non-supplemented and supplemented.

(Prediction 5b) than the supplemented group. Results for anterior tooth size are presented in Table 6. Results for trait frequency of tuberculum dentale are presented in Table 7. Compared to the supplemented group, the non-supplemented group, showed strong evidence (0.01>p>0.001) for smaller central incisor and canine crown sizes, while we found moderate to strong (0.05>p>0.001) evidence for larger lateral incisor crown sizes in the non-supplemented group. For trait expression, compared to the supplemented group, the non-supplemented group showed very strong evidence (p<0.001) for reduced frequency of the basally located tuberculum dentale in central incisors and strong evidence in canines. Results for lateral incisors showed weak support (0.1>p>0.05) for a *decrease* in tuberculum dentale trait expression in the non-supplemented group.

## Hypothesis 6: Molar trait expression and the Patterning Cascade Model in the entire sample

Hypothesis 6 predicts that peripheral accessory traits (Carabelli) and central accessory traits (upper cusp 5) may form under different conditions but are dependent on the spacing of principal molar cusps, as summarized in Fig 1. To address the degree to which the pattern of accessory cusp expression conforms to PCM expectations, we combined the entire sample to compare principal cusp distances and crown sizes of molars with and without the aforementioned traits. Results for Carabelli trait are presented in Table 8, and results for cusp 5 trait are reported in Table 9.

Comparing molars with Carabelli trait to molars without this trait, we found moderate evidence (0.05>p>0.01) for an increase in absolute spacing of paracone-protocone (ICD1) and paracone-hypocone (ICD5) distances, and moderate to strong (0.05>p>0.001) evidence for an increase in metacone-hypocone (ICD3) distances. For relative principal cusp spacing in molars with the Carabelli trait, we found strong to very strong evidence (p<0.001) for a decrease in the paracone-metacone (ICD2) dimension, moderate evidence (0.05>p>0.01) for a decrease in the protocone-hypocone (ICD4) dimension, and moderate to strong (0.05>p>0.001) evidence for a decrease in the metacone-protocone (ICD6) dimension. Comparing crown size for molars with and without the Carabelli trait, we found very strong support (p<0.001) for an increase in crown area for molars with the Carabelli trait present.

Comparing molars with cusp 5 trait to molars without this trait, results showed strong evidence (p<0.001) for a decrease in absolute principal cusp spacing of paracone-metacone (ICD2) and protocone-hypocone (ICD4) distances, moderate to strong evidence (0.05>p>0.001) of a decrease in spacing of paracone-hypocone (ICD5) distances, and

**Table 6. Wilcoxon test results showing differences in crown sizes of central incisors (I1), lateral incisors (I2), and canines (C) between the supplemented and non-supplemented groups.**

| | | | | | Comparing Groups | | | |
|---|---|---|---|---|---|---|---|---|
| Tooth type | W | df | *p*-value | Supplemented Mean Area | Non-supplemented Mean area | Differences in mean areas (supplemental–non-supplemental) | 95% Confidence Interval | Effect Size* |
| I1 | 0 | 1 | <0.001 | 64.510 | 45.439 | 19.071 | -0.114; 0.221 | 0.0330 (very small) |
| I2 | 0 | 1 | 0.018 | 48.850 | 62.699 | -13.849 | -0.016; 0.375 | 0.152 (small) |
| C | 0 | 1 | 0.010 | 68.183 | 64.718 | 3.465 | 0.077; 0.409 | 0.243 (small) |

*Effect size magnitude based on Cohen [56] where |r| <0.1: very small effect; |r| = 0.1: small effect; |r| = 0.3: medium effect, and |r| = 0.5: large effect

**Table 7. Wilcoxon test results showing differences in frequency of tuberculum dentale trait for central incisors (I1), lateral incisors (I2), and canines (C) between the supplemented and non-supplemented groups and Cliff's Delta effect size.**

| Trait | Tooth Type | W | df | p-value | 95% Confidence Interval | delta* |
|---|---|---|---|---|---|---|
| | | | | **Comparing Groups** | | |
| Tuberculum Dentale | I1 | 1955 | 1 | 0.001 | 0.1356–0.4702 | 0.3125 (small) |
| | I2 | 1567.5 | 1 | 0.064 | 0.0429–0.3980 | 0.2280 (small) |
| | C | 1742 | 1 | 0.010 | 0.0687–0.4383 | 0.2632 (small) |

*delta magnitude based on thresholds outlined by Romano et al. [57] where |delta| <0.146 is a negligible effect; |delta| <0.33 is a small effect; |delta| < 0.474 is a medium effect, and >0.474 is a large effect

Note: statistical results presented here are based on data shown in S2

moderate evidence (0.05>p>0.01) for a decrease in metacone-protocone (ICD6) distances. For relative principal cusp spacing of molars with cusp 5 trait, results showed moderate evidence (0.05>p>0.01) for a decrease in paracone-metacone (ICD2), protocone-hypocone

**Table 8. Wilcoxon test results showing differences in absolute intercusp distances, relative intercusp distances, and crown size between molars without and with Carabelli trait.**

| | | W | df | p-value | Average distance: Carabelli trait absent | Average distance: Carabelli trait present | Difference in average distances (absent–present) | 95% Confidence Interval | Effect Size* |
|---|---|---|---|---|---|---|---|---|---|
| Absolute ICD[1] | ICD1 | 739.5 | 1 | 0.026 | 7.490 | 7.763 | -0.273 | -0.459; -0.022 | 0.227 (small) |
| | ICD2 | 1153.5 | 1 | 0.309 | 5.577 | 5.461 | 0.116 | -0.131; 0.374 | 0.105 (small) |
| | ICD3 | 589.5 | 1 | 0.014 | 6.613 | 6.890 | -0.277 | -0.542; -0.068 | 0.260 (small) |
| | ICD4 | 1032 | 1 | 0.947 | 5.287 | 5.160 | 0.127 | -0.256; 0.287 | 0.00722 (very small) |
| | ICD5 | 610 | 1 | 0.023 | 9.352 | 9.721 | -0.369 | -0.645; -0.038 | 0.241 (small) |
| | ICD6 | 770 | 1 | 0.383 | 8.043 | 8.153 | -0.110 | -0.394; 0.156 | 0.0928 (very small) |
| Relative ICD[1] | ICD1 | 1187 | 1 | 0.201 | 0.727 | 0.714 | 0.013 | -0.007; 0.033 | 0.132 (small) |
| | ICD2 | 1437 | 1 | 0.001 | 0.541 | 0.503 | 0.038 | 0.015; 0.062 | 0.332 (small) |
| | ICD3 | 984 | 1 | 0.320 | 0.641 | 0.630 | 0.011 | -0.013; 0.033 | 0.106 (small) |
| | ICD4 | 1316 | 1 | 0.022 | 0.513 | 0.476 | 0.035 | 0.003; 0.047 | 0.235 (small) |
| | ICD5 | 1042 | 1 | 0.133 | 0.908 | 0.890 | 0.018 | -0.004; 0.037 | 0.160 (small) |
| | ICD6 | 1148 | 1 | 0.015 | 0.781 | 0.746 | 0.035 | 0.005; 0.051 | 0.258 (small) |
| Crown Size | Crown Area | 501 | 1 | <0.001 | 106.740 | 118.761 | -12.021 | -0.834; -0.328 | 0.419 (medium) |

[1]"ICD" = Intercusp Distance. Refer to Fig 2 for distances.

*Effect size magnitude based on Cohen [56] where |r| <0.1: very small effect; |r| = 0.1: small effect; |r| = 0.3: medium effect, and |r| = 0.5: large effect

**Table 9. Wilcoxon test results showing differences in absolute intercusp distances, relative intercusp distances, and crown size between molars without and with cusp 5 trait.**

| | | W | df | _p_-value | Average distance: Cusp 5 trait absent | Average distance: Cusp 5 trait present | Difference in average distances (absent–present) | 95% Confidence Interval | Effect Size* |
|---|---|---|---|---|---|---|---|---|---|
| Absolute ICD[1] | ICD1 | 1246.5 | 1 | 0.130 | 7.73 | 7.55 | 0.180 | -0.055; 0.379 | 0.156 (small) |
| | ICD2 | 1419.5 | 1 | 0.004 | 5.627 | 5.287 | 0.340 | 0.114; 0.609 | 0.292 (small) |
| | ICD3 | 964 | 1 | 0.809 | 6.815 | 6.775 | 0.040 | -0.208; 0.269 | 0.0260 (very small) |
| | ICD4 | 1410 | 1 | 0.005 | 5.334 | 4.981 | 0.353 | 0.103; 0.537 | 0.285 (small) |
| | ICD5 | 1225.5 | 1 | 0.014 | 9.724 | 9.401 | 0.323 | 0.067; 0.602 | 0.260 (small) |
| | ICD6 | 1187 | 1 | 0.033 | 8.205 | 7.993 | 0.212 | 0.015; 0.479 | 0.226 (small) |
| Relative ICD[1] | ICD1 | 1250 | 1 | 0.123 | 0.725 | 0.708 | 0.017 | -0.004; 0.035 | 0.158 (small) |
| | ICD2 | 1339 | 1 | 0.026 | 0.527 | 0.496 | 0.031 | 0.003; 0.056 | 0.229 (small) |
| | ICD3 | 965 | 1 | 0.803 | 0.636 | 0.631 | 0.005 | -0.018; 0.023 | 0.0269 (very small) |
| | ICD4 | 1338 | 1 | 0.026 | 0.501 | 0.469 | 0.032 | 0.002; 0.048 | 0.228 (small) |
| | ICD5 | 1174 | 1 | 0.044 | 0.908 | 0.876 | 0.032 | 0.008; 0.044 | 0.214 (small) |
| | ICD6 | 1137 | 1 | 0.088 | 0.766 | 0.745 | 0.021 | -0.002; 0.037 | 0.181 (small) |
| Crown size | Crown Area | 1047 | 1 | 0.088 | 114.710 | 114.372 | 0.338 | -0.290; 0.270 | 0.00237 (very small) |

[1]"ICD" = Intercusp Distance. Refer to Fig 2 for distances.

*Effect size magnitude based on Cohen [56] where |r| <0.1: very small effect; |r| = 0.1: small effect; |r| = 0.3: medium effect, and |r| = 0.5: large effect

(ICD4), and paracone-hypocone (ICD5) dimensions, and weak evidence (0.1>p>0.05) for a decrease in metacone-protocone (ICD6) dimensions. Comparing crown size for molars with and without the cusp 5 trait, we found weak evidence (0.1>p>0.05) for a difference in crown area between the two groups.

## Discussion

Our findings support the growing body of literature [18, 19, 59] that epigenetic factors, such as nutritional deprivation, can affect tooth size and morphology. Differences in the expression of crown size, molar cusp spacing, and dental trait expression between the supplemented and non-supplemented groups of Tezonteopan suggest that mild to moderate malnutrition can affect some aspects of tooth morphogenesis. The nutritional differences experienced by these groups did not result in gross morphological differences, as is the case in some experimental studies of extreme nutritional deprivation in non-human animals [20, 28, 29]. However, the findings presented here show that, in this sample, nutritional deprivation experienced during tooth morphogenesis: 1) is not related to molar crown size, 2) is associated with the spacing of

some principal cusp pairs and the expression of some accessory traits, and 3) appears to differentially affect anterior tooth size compared to molar size (Table 10). With respect to PCM expectations, findings were unclear in the comparison of supplemented to non-supplemented groups as outlined in hypotheses 1–4. However, comparing molars with and without Carabelli and cusp 5 in the entire sample allowed us to more directly investigate whether cusp configurations were associated with accessory cusp presence. For this comparison we found that principal cusp configurations and crown size largely conformed to PCM expectations for accessory cusp formation.

## Molar crown size and dietary stress

Our results for molar crown size do not support the prediction of smaller teeth in the non-supplemented group. We found little to no evidence that the first molars were smaller in the non-supplemented group, and only weak evidence that second molars were smaller. The power to detect differences may have been affected by our small sample sizes and the expected small effect size associated with crown size. It is also possible that first molars, which initiate around 30 weeks of gestation and are the only permanent tooth initiated *in utero* [60], may be more buffered (canalized) than later-forming teeth such as the second molar.

Another possibility for the lack of evidence of smaller molars in the non-supplemented group is that the caloric deficiency of the non-supplemented group might not have reached levels of severe deprivation that have been induced experimentally in animal studies such as Tonge and McCance's [20] study on pigs. In general, Tezonteopan residents consumed an average of 725 calories per day and 19.2 grams of protein, which is less than the daily recommended quantities of 1,250 calories and 32g [26:17]. Individuals in the supplemented group

**Table 10. Proposed predictions and associated support based on presented data.**

| Hypothesis (number) and Prediction (letter) | Strength of Evidence |
|---|---|
| 1: Non-supplemented group will have smaller molars | Little to no evidence of a difference between groups. |
| 2: Non-supplemented group will show smaller absolute and relative intercusp distances | Small but consistent differences* in cusp spacing (sign test) in the predicted direction. |
| 3(a): Non-supplemented group will show higher frequency of Carabelli cusp | Little to no evidence of higher frequency in predicted direction. |
| 3(b): Non-supplemented group will have greater Carabelli cusp scores | Moderate evidence of positive overall effect of non-supplementation on Carabelli cusp scores |
| 4(a): Non-supplemented group will show greater frequency of upper cusp 5 | Moderate evidence of greater frequency in the predicted direction. |
| 4(b): Non-supplemented group will show greater scores of upper cusp 5 | Strong evidence of positive overall effect of non-supplementation on cusp 5 score categories |
| 5(a): Non-supplemented group will show smaller anterior teeth | Strong evidence for smaller crown sizes of central incisors and canines in the predicted direction. Strong evidence for larger lateral incisors contrary to the predicted direction. |
| 5(b): Non-supplemented group will show lower frequency of tuberculum dentale | Strong evidence of lower frequency in central incisors and canines in the predicted direction. Weak evidence of opposite trend for lateral incisor. |
| 6: Carabelli trait will be present when conditions associated with Fig 1 configurations B, C or D are realized | Moderate to strong evidence for conditions associated with two predicted configurations |
| 6: Cusp 5 will be present when conditions associated with Fig 1 configurations B, C or D are realized | Moderate to strong evidence for conditions associated with one predicted configuration |

*Here we refer to differences in means between supplemented and non-supplemented groups for absolute and relative intercusp distances (see Table 2).

were given fortified milk 2–3 times per day, which brought their dietary intake of calories and protein closer to recommended levels. Non-supplemented individuals in the present study had, in addition to a comparatively poor diet, more frequent and longer bouts of sickness, and reduced body mass compared to individuals in the supplemented group [26]. Infants born to mothers in the supplemented group had greater birth weights and lost less weight just after birth compared to non-supplemented-group infants, suggesting that individuals in the supplemented group were better nourished than the non-supplemented individuals [26]. Height and weight comparisons from 18 months to two years of age showed individuals in the non-supplemented group weighed less and exhibited shorter stature than their supplemented counterparts [26]. Thus, although the two groups differed in nutritional intake, in body weight, and in growth, the dietary differences between them may not have been great enough to affect molar crown size.

## Molar intercusp distance and dietary stress

The prediction that molar principal cusp distances will be more affected by nutritional deprivation than tooth crown dimensions is generally supported by our results. Evidence of smaller diagonal dimensions, especially between the metacone and protocone (ICD6), and a directional effect towards reduced principal intercusp spaces in the non-supplemented group suggest that cusp spacing may be affected by dietary intake during the development period. These findings are consistent with the study by Townsend and colleagues [18] suggesting that the placement of and space between cusps are more affected by epigenetic factors during morphogenesis than are crown dimensions. Different morphogenic processes are likely responsible for principal cusp spacing and crown size. Configuration and timing of secondary enamel knots affects cusp spacing while growth rates and duration affect crown size.

## Carabelli trait expression and dietary stress

Carabelli trait frequencies were not different between the supplemented and non-supplemented groups, but the non-supplemented group had a tendency toward greater expression of Carabelli trait. Therefore, we will focus on Carabelli expression differences. Although there was no evidence of smaller molar crown sizes in the non-supplemented group, the directional decrease of small magnitude in nearly all principal cusp spacings may have provided conditions that favored an increase in Carabelli trait expression within the non-supplemented group. These conditions are consistent with Fig 1 Configuration B, first proposed by Hunter and colleagues for Carabelli trait presence [22], in which all principal cusps are closer together. Given that the trait expression data, but not the frequency data, are aligned with PCM expectations, our results might be consistent with the hypothesis Riga et al. [19] proffered: that nutritional stress introduces instability in crown formation making outcomes based on the PCM less predictable. Exploring this possibility in a larger sample may elucidate the relationship between crown characteristics and nutritional stress.

## Cusp 5 expression and dietary stress

The higher frequency and presence of elevated expressions of the upper cusp 5 trait (here found only on M1s) in the non-supplemented group supports our Predictions 5a and 5b, namely that individuals experiencing nutritional deprivation will show higher frequencies of and greater ASUDAS scores for this trait. We found a small but directional decrease in all absolute and most relative principal cusp distances in the non-supplemented group compared to the supplemented group. If we consider cusp 5 a central cusp situated directly between the metacone and hypocone, these conditions do not directly match with the proposed

configurations associated with the presence of the cusp 5 trait (Fig 1 Configurations A, C and/ or D). But, considering that the placement of the cusp 5 trait is more distal than either of the nearby principal cusps, this accessory trait is as much a central cusp as it is a peripheral cusp. Keeping crown size constant, it is possible that conditions with reduced mesiodistal, buccolingual, and diagonal principal cusp dimensions could favor the formation of the cusp 5 trait.

### Anterior dentition and dietary stress

As predicted, we found smaller central incisors and canines in the non-supplemented group compared to the supplemented group. These results differ from those for the molars where evidence for size differences between groups was minimal. This suggest that crown size may respond differently to adequate nutritional intake in anterior as opposed to posterior teeth. The independent nature of these two tooth groups during development is consistent with research by Hlusko and colleagues [61] who found that phenotypic variation of incisor and molar crown sizes of mice and baboons were genetically independent of each other, suggesting separate developmental modules for incisors and molars during embryogenesis.

Evidence from lateral incisors did not conform to the prediction, as the non-supplemented group showed weak evidence of larger crown sizes for this tooth type than the supplemented group. Different populations have different calcification initiation times for these teeth, but according to the London Atlas of Dental Development, upper second incisors initiate calcification approximately 7 months after central incisors and approximately 3 months after canines [62]. Perhaps these differences in developmental timing account for the differences among these anterior teeth.

As predicted, anterior teeth of the non-supplemented group exhibited lower frequency of the tuberculum dentale in central incisors and canines compared to the supplemented group. We expected the larger anterior teeth of the supplemented group to have more basal features possibly as a direct result of greater mesenchymal proliferation [32, 34]. Enhanced growth may also indirectly increase the likelihood of forming a secondary enamel knot responsible for the variable features as the dental epithelium continues to grow further downward and outward [25]. It seems possible that secondary enamel knots are involved in the morphogenesis of anterior teeth as well as molars, based on observed incisor complexity in some mammals (e.g., the three-cusped upper central incisors of plesiadapids; [see 63] and the rare occurrence of talon cusps in humans [64–66] and baboons [67]). Basally located traits like the talon cusp and tuberculum dentale could be regulated by such secondary enamel knots.

### Accessory cusps and the Patterning Cascade Model in the entire sample

Similar to previous studies [18, 22, 23], in the present study both molar crown size and principal cusp spacing were associated with the presence of Carabelli and cusp 5. For Carabelli trait, there was an increase in the absolute values of the buccolingual spacing of the paracone-protocone and metacone-hypocone, and the diagonal distance of the paracone-hypocone. These increases in dimensions are not consistent with any of the four configurations presented in Fig 1. However, when the larger crown size of molars with Carabelli trait is taken into consideration, these increased intercuspal distances result in decreases in relative principal cusp spacing.

Relative intercusp distances were more consistent with the configurations summarized in Fig 1. Here, we found evidence for a decrease in the mesiodistal dimensions of the paracone-metacone and protocone-hypocone, as well as a diagonal decrease in the paracone-hypocone dimension. These conditions follow Fig 1 Configuration D in which mesiodistal decreases in the spacing of upper molar principal cusps are suggested to promote the formation of the

Carabelli trait. The decrease in the diagonal dimension also suggests a slight change in principal cusp arrangement.

Evidence from cusp 5 was consistent with PCM expectations as visualized in Fig 1. For both absolute and relative intercusp distances, we found a decrease in the mesiodistal dimensions (between paracone-metacone and protocone-hypocone) as well as a decrease in the diagonal dimensions (between paracone-hypocone and to a lesser degree the metacone-protocone) of the principal cusps. Crown size was found to be similar between molars with and without the cusp 5 trait. These conditions are consistent with Fig 1 Configuration D wherein a decrease in mesiodistal cusp dimensions favors the formation of the cusp 5 trait. Given its placement near the metacone and hypocone, upper cusp 5 expression is likely sensitive to its overall proximity to the distal main cusps (metacone and hypocone) [17, 23], which in our sample are closer to their mesial counterparts (paracone and protocone, respectively) on molars with cusp 5. If we consider cusp 5 to be both a central as well as a peripheral accessory trait, it could be that the decrease in mesiodistal as well as diagonal dimensions of the principal cusp spacings may also favor the formation of this trait.

## Conclusions

This study revisits the long-held assumption that dental morphological traits are not affected by developmental environments. Findings presented here suggest that nutritional deprivation during tooth morphogenesis may contribute to changes in the phenotype of tooth characteristics and may affect the anterior and posterior dentition differently. These findings are also consistent with Townsend et al.'s [18] conclusion of greater epigenetic influence on principal cusp spacing than on crown size. Differences in tooth size and cusp expression between the supplemented and non-supplemented groups suggests that nutritional stress can influence morphological outcomes, as suggested by Riga et al. [19]. In the present study, conclusions related to expectations of the PCM were twofold. First, one configuration (Fig 1D) better fits our data than other configurations (Fig 1A–1C), but that may not be the case for other samples. Second, our results were more consistent with the PCM when comparing molars with and without traits in the whole sample, than for comparisons between the supplemented and non-supplemented groups. Thus, the PCM, while effective at explaining morphological differences in general, may be less successful at explaining phenotypic changes under conditions of stress. Overall, the findings presented here are consistent with others showing that developmental physiological stress can alter dental phenotypes [19, 20, 36, 37, 68]. Although the effects of cusp distances and crown size are small, as shown here and elsewhere [22], they may be worth considering and should be tested on a larger sample. Given the broad application of dental characteristics in biodistance studies [1–7], it may also be useful to consider the impact that developmental stress might have on these traits.

## Supporting information

**S1 Table. Results from Shapiro-Wilks tests for normality.** Note: All teeth are uppers. M1 = first molar, M2 = second molar, I1 = first incisor, I2 = second incisor, C = canine. [1]ICD = intercusp distance. See Fig 2 for specific distances.
(DOCX)

**S2 Table. Comparison of crown traits between the supplemented and non-supplemented groups.** Note: Breakpoints were set following Hanihara [45] and Scott et al. [42].
(DOCX)

## Acknowledgments

Thank you to the journal editors and anonymous reviewers for feedback on drafts of this manuscript. We would like to acknowledge the study participants and the directors, Dr. Adolfo Chavez and Celia Martinez. Thank you to JG and TMS for reading early drafts and providing feedback.

## Author Contributions

**Conceptualization:** Erin C. Blankenship-Sefczek, John P. Hunter, Debbie Guatelli-Steinberg.

**Data curation:** Erin C. Blankenship-Sefczek.

**Formal analysis:** Erin C. Blankenship-Sefczek, Debbie Guatelli-Steinberg.

**Investigation:** Erin C. Blankenship-Sefczek.

**Methodology:** Erin C. Blankenship-Sefczek, Debbie Guatelli-Steinberg.

**Resources:** Alan H. Goodman.

**Validation:** Erin C. Blankenship-Sefczek.

**Writing – original draft:** Erin C. Blankenship-Sefczek.

**Writing – review & editing:** Erin C. Blankenship-Sefczek, Alan H. Goodman, Mark Hubbe, John P. Hunter, Debbie Guatelli-Steinberg.

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
