## [Decision Letter · Decision Letter 0]

26 Jun 2023

PONE-D-23-12197Nutritional supplementation, tooth crown size, and trait expression in individuals from Tezonteopan, Mexico.PLOS ONE

Dear Dr. Blankenship-Sefczek,

Thank you for submitting your manuscript to PLOS ONE. After careful consideration, we feel that it has merit but does not fully meet PLOS ONE’s publication criteria as it currently stands. Therefore, we invite you to submit a revised version of the manuscript that addresses the points raised during the review process. I now have two expert reviews of this work. There are significant concerns over the structure of the manuscript, rationale, statistical approach, and discussion of results that may not reflect what the authors were able to demonstrate with the limited data set.

We look forward to receiving your revised manuscript.

Kind regards,

JJ Cray Jr., Ph.D.

Academic Editor

PLOS ONE

2. We note that Figures 2 and 3 in your submission contain copyrighted images. All PLOS content is published under the Creative Commons Attribution License (CC BY 4.0), which means that the manuscript, images, and Supporting Information files will be freely available online, and any third party is permitted to access, download, copy, distribute, and use these materials in any way, even commercially, with proper attribution. For more information, see our copyright guidelines: http://journals.plos.org/plosone/s/licenses-and-copyright.

a. You may seek permission from the original copyright holder of Figures 2 and 3 to publish the content specifically under the CC BY 4.0 license.

b.If you are unable to obtain permission from the original copyright holder to publish these figures under the CC BY 4.0 license or if the copyright holder’s requirements are incompatible with the CC BY 4.0 license, please either i) remove the figure or ii) supply a replacement figure that complies with the CC BY 4.0 license. Please check copyright information on all replacement figures and update the figure caption with source information. If applicable, please specify in the figure caption text when a figure is similar but not identical to the original image and is therefore for illustrative purposes only.

Reviewers' comments:

Reviewer's Responses to Questions

**Comments to the Author**

1. Is the manuscript technically sound, and do the data support the conclusions?

Reviewer #1: Yes

Reviewer #2: Partly

2. Has the statistical analysis been performed appropriately and rigorously? 

Reviewer #1: Yes

Reviewer #2: I Don't Know

3. Have the authors made all data underlying the findings in their manuscript fully available?

Reviewer #1: Yes

Reviewer #2: Yes

4. Is the manuscript presented in an intelligible fashion and written in standard English?

Reviewer #1: Yes

Reviewer #2: Yes

5. Review Comments to the Author

Reviewer #1: Thank you for giving me the opportunity to review this article. In this manuscript, the authors explored the relationship between nutritional stress and tooth size, intercusp spacing, and morphological trait presence and expression. This work is exciting since to date, it is the first to investigate how environmental factors may influence the degree to which expectations under the patterning cascade model of tooth morphogenesis are followed. The authors found no support that nutritional stress results in smaller molars and higher frequencies of Carabelli’s trait. They, however, did find partial support that significant differences exist in intercusp spacing, Carabelli’s cusp trait expression, cusp 5 presence and expression, anterior tooth size and trait expression between nutritionally supplemented and non-supplemented groups. Additionally, the authors found that the spacing between the lingual cusps and buccal cusps of the upper molars appear to be the significant predictors of accessory cusp development. While the results of the research may be tempered by the small sample size, the authors do acknowledge this throughout the manuscript and the value provided by the results of this novel analysis far outweigh sample size concerns. Below, I provide specific comments that I hope will improve the manuscript:

Line 77-80: The way this sentence is currently worded is confusing. Suggested rephrase: The PCM has emerged as a developmental framework to understand how enamel knot spacing and the space and time available for crown formation affect variation in dental morphology.

Line 117-119: The sample of citation 16 was actually conducted on modern humans, I suggest this citation be placed in the appropriate section within the sentence. Additionally, some of the evidence found in the cited studies, especially those that were conducted in fossil hominins and modern humans, provides mixed support for the PCM. I would incorporate this into the sentence to indicate that previous work has not found unequivocal support for this model, especially as it pertains to humans, which is relevant for this study since the current study’s sample is composed of modern humans. This will also bolster this study’s results since this research, similar to previous ones conducted in humans, only found partial support for the PCM.

Line 148: For Hypothesis 2, do the authors expect to find any specific trends with respect to cusp spacing? Would the authors expect to see smaller intercusp distances or larger ones in supplemented vs. non-supplemented groups? The authors provide concrete trends they expect for the other hypotheses based on previous research, whereas in Hypothesis 2, their prediction seems to indicate that they just expect the two groups to exhibit significant differences. If the authors expect specific trends, it would be helpful to clarify here what those would be. If they are unsure about any trends (maybe because no prior study has examined this) that might also be helpful to clarify here. The authors also discuss downstream changes they expect to see in intercusp spacing in hypothesis 3 and 4 as it relates to accessory cusp formation. If these are the specific expectations the authors have for changes in intercusp distance, it might be helpful to reference hypothesis 3 and 4 for the specific trends. For example: “See Hypothesis 3 and 4 for trends expected for intercusp spacing under the PCM”.

Lines 213- 276: The Study Materials section is fairly long. In my opinion, the most important parts are the first and last paragraph of this section. The information provided in the paragraph beginning on line 235 could be reduced and incorporated into the paragraph preceding it. The study by Goodman et al. discussed in the paragraph starting on line 244 would fit well in the discussion around environmental stress and dental phenotypes of the Background section (lines 123-135).

The information provided in the sentences located on lines 257-258, lines 268-270, and lines 270-272 is identical and feels repetitive. This section would read better if some of this overlapping information was reduced.

Lines 278-279: A citation indicating the methods that were followed to obtain the BL and MD measurements would be beneficial here.

Line 322: The authors indicate that trait frequency based on trait presence and absence was compared between supplemented and non-supplemented groups. However, the authors only discuss data collection methodology for raw scores in the previous paragraph (lines 300-303). In that section (line 304), it would be helpful to add a sentence discussing how traits were dichotomized to presence/absence states and citing the system that was followed for dichotomization.

General comment for Tables 3, 4, and 6: The way dental morphological trait frequency is usually presented is by providing trait frequency and the total number of individuals for which the trait was observable. The way they are currently presented is confusing. Are the numbers below “Supplemented” and “Non-supplemented” referring to the number of individuals who were characterized by the respective traits? Are they the % frequency of the trait? I would recommend providing the frequency of the trait (as a proportion) followed by the total number of individuals for whom observations could be made (n), especially since the Wilcoxon test is used to test for difference between the frequencies. If the authors need further clarification on what is meant by the comment here, please refer to Table 3 in Irish, 1997 for how morphological trait frequencies are usually presented, which is a publication cited within this manuscript.

Lines 391-398: The authors identified a significant difference in cusp 5 frequency and expression, but this was only so for M1, not M2. This should be made clear in this section.

Line 396: The authors argue that the results of their analyses indicate that the non-supplemented group had higher grades of cusp 5 expression, but information supporting this is not presented in the manuscript. Table 3, cited to present this information, only indicates results for when dichotomized trait presence is compared between the groups, but not expression. It would be helpful if the authors presented these results in a table, similar to how these results were presented for Carabelli’s trait in Table 4.

Lines 404-408: This section has several statements that do not align with the results presented in Table 5. The authors state that “The supplemented group exhibited significantly larger central incisors (Wilcoxon test: W=2470.5, df=1, p=0.018) and canines (Wilcoxon test: W=1742, df=1, p=0.010), compared to the non-supplemented group”. However, the results attributed to the central incisor in this statement are actually those of the lateral incisor (based on Table 5). While central incisors are larger in the supplemented group, these results are not significant (p=0.404). Conversely, the following sentence discussing the results for the lateral incisor cites Table 6, whereas the results are actually presented in Table 5. Please also correct the sentence in lines 408-410 to refer to the appropriate results as well.

Lines 413-416: While the central incisors and canines of the supplemented group did exhibit significantly higher frequencies of tuberculum dentale than the non-supplemented groups, this was not the case for the lateral incisor, which arguably provides mixed support for prediction 5b, rather than complete support.

Lines 438-439: Authors mixed up the two cusps here. Carabelli’s trait would be a peripheral accessory cusp, while cusp 5 would be a central accessory cusp.

Lines 436-456: To test hypothesis 6, it might be beneficial to run analyses separately for Carabelli’s trait and cusp 5, rather than lumping them together as 5+ cusped molars. This could be presented as an additional table to Table 6 or even as a supplementary table. As the authors point out, Carabelli’s trait is a peripheral accessory cusp, while cusp 5 is a central accessory cusp. Due to differential placement on the crown, some of the intercusp distance expectations would be different for these two cusps. For Carabelli’s all intercusp distances should be smaller to support the potential of accessory cusp development on the periphery (especially that between cusp 1 and 2). On the other hand, to ensure that a centrally placed accessory cusp 5 can develop, we would expect increased distance between cusps 3 and 4 to ensure the fields of inhibition surrounding these cusps allow additional cusp development. This contradicts predictions for Carabelli’s trait. Therefore, I think it would be helpful to show those analyses separately to make sure that considering the two traits at the same time isn’t obscuring the results obtained.

Table 7: As with the previous tables, what is located below the “Supplemented” and “Non-supplemented” columns is very ambiguous. I assume it’s the average intercusp distance within each group? This should be made clear in the labels. For example, instead of simply saying “Supplemented”, I recommend changing the label to “Average ICD (intercusp distance) for supplemented group”.

Table 8: The authors assert that Hypothesis 2 was supported because significant differences between the supplemented and non-supplemented group were found for intercusp spacing. However, these results were only significant for the protocone-metacone distance. Wouldn’t this only lend partial support to Hypothesis 2? Similarly, for Hypotheses 4a and 4b, the results were only significant for M1, but not M2. Wouldn’t this lend partial support to the hypotheses? This is also the case for Hypotheses 5a and 5b as not all results were significant.

Lines 503-504: Again, since these differences are only significant for the protocone-metacone distance, wouldn’t this hypothesis be partially supported?

Lines 534-535: As mentioned previously, differences in central incisor size while larger in the supplemented group, these results are not significant. However, the lateral incisor seems to be significantly larger in the non-supplemented group.

Reviewer #2: Thank you for the opportunity to review this interesting manuscript, “Nutritional supplementation, tooth crown size, and trait expression in individuals from Tezonteopan, Mexico”. This paper employs a dataset collected in Mexico between 1969-1980 to attempt to address questions related to the role of dietary/environmental stress on tooth morphogenesis in humans. The authors focus on the maxillary permanent dentition, and analyze data from dental casts collected during the original study. In the original study, half of participant mothers were given nutritional supplementation and half were not. In the current sample (n=73), n=34 were children of mothers who received nutritional supplementation. Overall, this is an interesting study and fairly unique dataset, but the strength of the data is weak given the attempt to use this particular sample to draw more generalizable conclusions. I would strongly suggest the authors should reframe this as a story about this very unique sample and then speak in more measured terms about the potential applicability to broader concepts such as the PCM.

-This manuscript is quite long. There are two whole pages outlining the hypotheses, after listing research aims. Strongly suggest reducing and synthesizing this section, with perhaps a bulleted list of hypotheses at the end. Or just synthesize and include (brief) hypothesis statements in the results section since these are already briefly reiterated there. There are likely other areas that could be trimmed, particularly in the intro/background.

-The authors engage in multiple testing but not correction of their p-value cut-off. Every hypothesis you add that uses the same underlying sample increases the odds of a false positive. Given this and the small sample size (and thus likely limited power), the authors should be more cautious about not over-interpreting ‘significant’ results. I am concerned about the applicability of their findings to humans more broadly. This is an interesting story about dental morphology in this group, but I am concerned the small sample size and inability to control for confounders may mean these results are not reproducible in other samples.

-The authors have a near-equal number of supported and unsupported hypotheses, and many of their p-values are near the 0.05 cut-off (and would not make a more stringent cut-off correcting for the effects of multiple testing). It is difficult to therefore parse the broader significance of this work. I’m particularly not convinced that the authors have demonstrated “the PCM explains patterns of accessory cusp variation in the entire sample” given that three of the four sub-hypotheses were not supported by the data.

-I would suggest recontextualizing this as a story about this particular group and its relation to the dentition of the local population. What are the frequencies of the cusp traits/crown morphologies studied in this population as a whole? (By population I’m referring to the regional or ethnic group(s) represented in Tezonteopan). Which group (supplemented or unsupplemented) more closely mirrors the broader population in this area of Mexico? Please provide context for whether the supplemented or unsupplemented samples (or neither) is most representative of the predominant cultural group(s) (not specified in the paper) in this region. Is there any other such data from this region? If not, how does it compare to frequencies/tooth sizes for other groups in Mexico and/or the broader global average?

-The authors appear to be discussing the permanent dentition but do not explicitly state this (they do use the capital letter nomenclature used to denote permanent dentition, but please note that this is not commonly used in clinical dentistry). Suggest the authors explicitly state focus on permanent dentition at least once in the beginning of the paper

-If supplementation during pregnancy and weaning was the key aspect of diet, why focus exclusively on the permanent dentition? The relatively lengthy duration of breast feeding (2 years) still does not cover the full period of crown formation for the M2s. Are there dental casts which capture the deciduous dentition?

- I know LEH in the anterior teeth does not correspond to timeline of M1 but what about evidence of LEH in the M2s? It could be informative to note whether those with more LEH were also those who had smaller anterior teeth or were more likely to have accessory cusps

-Similarly, the authors indicate that the un-supplemented group had markers of growth perturbance, but do those who had stunting of their statural growth, for example, have more dental anomalies/smaller teeth? This (and/or LEH data) would help bolster the manuscript’s argument in light of the small sample size and borderline p-values for many of the analyses

-I understand how challenging it is to come by data like this, but given the limitations of the sample, this work would seem to fall more in the realm of hypothesis generation rather than rigorous hypothesis testing. If the authors were to expand their discussion of dental traits in the broader local population (or other regional Mexican populations) they might be able to make interesting observations about what might be similar/different in the presence of nutritional supplementation. This could then serve as a jumping off point for discussing how these results may hint at broader human patterns and indicate areas for future work.

6. PLOS authors have the option to publish the peer review history of their article (what does this mean?). If published, this will include your full peer review and any attached files.

Reviewer #1: No

Reviewer #2: No

---

## [Author Response · Author response to Decision Letter 0]

25 Aug 2023

With respect to our choices around statistical analyses, based on feedback from Reviewer 2, we reframed how we interpret our p-values, and expanded our explanation of the methodological and philosophical reasons for adopting particular statistical approaches. We recognize, as Reviewer 2 points out, that adjustments to p-values reduces the likelihood of Type I errors (false positives). Statistical tests run on small sample sizes with non-parametric tests, like ours though, also increase the likelihood of Type II errors (false negatives). Therefore, we chose not to correct for multiple tests. We understand that, given our small sample sizes, we should be more cautious in interpreting our “significant” results. Thus, we chose to follow the advice of Muff et al. 2022, using p-values to identify the relative strength of evidence rather than as binary cut-offs for “significant” or “not significant” results. We now qualify our p-values according to the schema these authors outlined: p-values greater than 0.1 imply "little or no evidence”, between 0.1 and 0.05 “weak evidence”, between 0.05 and 0.01 “moderate evidence”, between 0.01 and 0.001 “strong evidence”, and less than 0.001 “very strong evidence”. 

During the revision process, we realized that the ordinal regression for Carabelli trait expression was originally run as a two-tailed test. Given that the prediction was directional, however, we should have performed a one-tailed ordinal regression. The p-values associated with this test now reflect this correction in the manuscript. We added a new ordinal regression analysis for cusp 5 trait expression, and, as was the case for the Carabelli cusp, we performed a one-tailed test for the directional hypothesis of greater trait expression in the non-supplemented group relative to the supplemented group. These tests allow us to test the overall effect of supplementation on trait expression as well as specific associations between nutritional supplementation and transitions between trait scores. As a side note, we wish to point out that all of the original Wilcoxon tests in our study were run as one-tailed tests. 

We have made many additional changes that we believe strengthen the rigor of our manuscript. Below please find our detailed descriptions outlining how MS revisions were made in response to your feedback. 

Journal Requirements:

Comment 1: Please ensure that your manuscript meets PLOS ONE's style requirements, including those for file naming.

Response: The manuscript and associated files names have been edited following PLOS ONE’s style requirements.

Comment 2: We note that Figures 2 and 3 in your submission contain copyrighted images. All PLOS content is published under the Creative Commons Attribution License (CC BY 4.0), which means that the manuscript, images, and Supporting Information files will be freely available online, and any third party is permitted to access, download, copy, distribute, and use these materials in any way, even commercially, with proper attribution. We require you to either (1) present written permission from the copyright holder to publish these figures specifically under the CC BY 4.0 license, or (2) remove the figures from your submission.

Response: These images came from the first author’s dissertation published through ProQuest. ProQuest sent the following response to an inquiry about using copyright images from dissertations: “ProQuest’s author agreement for dissertations and theses is non-exclusive. Authors have the full right to make their works available to other commercial services or for open access outside of the ProQuest service.” The requested permission form has been filled out by the first author and is included as one of the resubmission files. Since the first author is the original copyright holder, ProQuest does not provide additional permission for distribution. 

Reviewer 1:

Comment 1: Line 77-80: The way this sentence is currently worded is confusing. Suggested rephrase: The PCM has emerged as a developmental framework to understand how enamel knot spacing and the space and time available for crown formation affect variation in dental morphology.

Response: Line 63-: The sentence was changed as follows: The PCM has emerged as a powerful conceptual framework for understanding how enamel knot spacing and the space available for crown formation affect variation in dental morphology [17,21-24].

Comment 2: Line 117-119: The sample of citation 16 was actually conducted on modern humans, I suggest this citation be placed in the appropriate section within the sentence. Additionally, some of the evidence found in the cited studies, especially those that were conducted in fossil hominins and modern humans, provides mixed support for the PCM. I would incorporate this into the sentence to indicate that previous work has not found unequivocal support for this model, especially as it pertains to humans, which is relevant for this study since the current study’s sample is composed of modern humans. This will also bolster this study’s results since this research, similar to previous ones conducted in humans, only found partial support for the PCM.

Response: Lines 91-95 were changed as follows: Studies have tested and supported different aspects of the PCM’s predictions for the dentitions of rodents [25,33,34] and seals [21], where timing and orientation of secondary enamel knots were used to predict placement of cusps and molar morphology. For application to quadritubercular molars, various principal cusp configurations have been proposed to increase the likelihood of forming accessory cusps [17, 22,23].

Comment 3: Line 148: For Hypothesis 2, do the authors expect to find any specific trends with respect to cusp spacing? Would the authors expect to see smaller intercusp distances or larger ones in supplemented vs. non-supplemented groups? The authors provide concrete trends they expect for the other hypotheses based on previous research, whereas in Hypothesis 2, their prediction seems to indicate that they just expect the two groups to exhibit significant differences. If the authors expect specific trends, it would be helpful to clarify here what those would be. If they are unsure about any trends (maybe because no prior study has examined this) that might also be helpful to clarify here. The authors also discuss downstream changes they expect to see in intercusp spacing in hypothesis 3 and 4 as it relates to accessory cusp formation. If these are the specific expectations the authors have for changes in intercusp distance, it might be helpful to reference hypothesis 3 and 4 for the specific trends. For example: “See Hypothesis 3 and 4 for trends expected for intercusp spacing under the PCM”.

Response: We thank Reviewer 1 for these questions, which helped us to think more clearly about our expectations for this hypothesis. Lines 161-167 were changed as follows: Hypothesis 2: Dietary stress interferes with the process of molar morphogenesis by altering the placement of the main molar cusps. In our study sample we expected that the non-supplemented group would have smaller absolute intercusp distances as well as smaller crown sizes (as per Hypothesis 1). Based on Townsend et al [18], intercusp distances are expected to be more affected by epigenetic factors than crown size. Hypothesis 2 thus predicts that the non-supplemented group will have both smaller absolute and relative principal intercusp distances compared to the supplemented group.

Comment 4: Lines 213- 276: The Study Materials section is fairly long. In my opinion, the most important parts are the first and last paragraph of this section. The information provided in the paragraph beginning on line 235 could be reduced and incorporated into the paragraph preceding it. The study by Goodman et al. discussed in the paragraph starting on line 244 would fit well in the discussion around environmental stress and dental phenotypes of the Background section (lines 123-135).

Response: As suggested, we reduced wording in some sections of the Study Collection section (Lines 204-256) and deleted the paragraph discussing the study by Goodman et al (1991) in relation to LEH. 

Comment 5: The information provided in the sentences located on lines 257-258, lines 268-270, and lines 270-272 is identical and feels repetitive. This section would read better if some of this overlapping information was reduced.

Response: This section was changed as follows: Lines 236-237: As part of their study, Goodman et al. [41] took dental impressions of dentition from the participants to document the presence of LEH. The second two lines were deleted.

Comment 6: Lines 278-279: A citation indicating the methods that were followed to obtain the BL and MD measurements would be beneficial here.

Response: Citation was added to line 258-259: Maximum buccolingual (BL) and mesiodistal (MD) lengths were recorded for maxillary dentitions using digital calipers [43; Moorrees and Reed 1964].

Comment 7: Line 322: The authors indicate that trait frequency based on trait presence and absence was compared between supplemented and non-supplemented groups. However, the authors only discuss data collection methodology for raw scores in the previous paragraph (lines 300-303). In that section (line 304), it would be helpful to add a sentence discussing how traits were dichotomized to presence/absence states and citing the system that was followed for dichotomization.

Response: Lines 287-291 were changed for clarity: For some statistical analyses, trait frequency (presence/absence) was compared. Compared to global averages, Indigenous Central and South American populations show lower frequencies of molar accessory cusps [45; Hanihara 2008]. Therefore, breakpoints for Carabelli’s trait and upper cusp 5 were set at 1+ [42; Scott et al. 2016].

Comment 8: General comment for Tables 3, 4, and 6: The way dental morphological trait frequency is usually presented is by providing trait frequency and the total number of individuals for which the trait was observable. The way they are currently presented is confusing. Are the numbers below “Supplemented” and “Non-supplemented” referring to the number of individuals who were characterized by the respective traits? Are they the % frequency of the trait? I would recommend providing the frequency of the trait (as a proportion) followed by the total number of individuals for whom observations could be made (n), especially since the Wilcoxon test is used to test for difference between the frequencies. If the authors need further clarification on what is meant by the comment here, please refer to Table 3 in Irish, 1997 for how morphological trait frequencies are usually presented, which is a publication cited within this manuscript.

Response: We thank Reviewer 1 for this comment. We added a supplemental table (S2 Table) with “n” and “%” for each tooth trait.

Comment 9: Lines 391-398: The authors identified a significant difference in cusp 5 frequency and expression, but this was only so for M1, not M2. This should be made clear in this section.

Response: Lines 397-399 were changed as follows: Compared to the supplemented group, the non-supplemented group showed moderate evidence for a greater frequency of cusp 5 trait presence in first molars. There were no cusp 5s present on second molars (M2s) in either group.

Comment 10: Line 396: The authors argue that the results of their analyses indicate that the non-supplemented group had higher grades of cusp 5 expression, but information supporting this is not presented in the manuscript. Table 3, cited to present this information, only indicates results for when dichotomized trait presence is compared between the groups, but not expression. It would be helpful if the authors presented these results in a table, similar to how these results were presented for Carabelli’s trait in Table 4.

Response: We thank Reviewer 1 for this suggestion. We performed and now include an Ordinal Logistic Regression analysis for cusp 5 expression, similar to the analysis that was done for the Carabelli trait. Results are outlined in lines 391-401 and in Table 5.

Comment 11: Lines 404-408: This section has several statements that do not align with the results presented in Table 5. The authors state that “The supplemented group exhibited significantly larger central incisors (Wilcoxon test: W=2470.5, df=1, p=0.018) and canines (Wilcoxon test: W=1742, df=1, p=0.010), compared to the non-supplemented group”. However, the results attributed to the central incisor in this statement are actually those of the lateral incisor (based on Table 5). While central incisors are larger in the supplemented group, these results are not significant (p=0.404). Conversely, the following sentence discussing the results for the lateral incisor cites Table 6, whereas the results are actually presented in Table 5. Please also correct the sentence in lines 408-410 to refer to the appropriate results as well.

Response: We thank Reviewer 1 for making this comment. Values in the table were originally entered incorrectly. These values have been changed to the correct values. Lines 415-418: Compared to the supplemented group, the non-supplemented group, showed strong evidence for smaller central incisor and canine crown sizes, while we found moderate to strong evidence for larger lateral incisor crown sizes in the non-supplemented group.

Comment 12: Lines 413-416: While the central incisors and canines of the supplemented group did exhibit significantly higher frequencies of tuberculum dentale than the non-supplemented groups, this was not the case for the lateral incisor, which arguably provides mixed support for prediction 5b, rather than complete support.

Response: We altered how we interpret our p-values. Lines 418-422 were changed as follows: For trait expression, compared to the supplemented group, the non-supplemented group showed very strong evidence for reduced frequency of the basally located tuberculum dentale in central incisors and strong evidence in canines. Results for lateral incisors showed weak support for a decrease in tuberculum dentale trait expression in the non-supplemented group.

Comment 13: Lines 438-439: Authors mixed up the two cusps here. Carabelli’s trait would be a peripheral accessory cusp, while cusp 5 would be a central accessory cusp.

Response: We thank Reviewer 2 for catching this. The change was made to lines 434-436: Hypothesis 6 predicts that peripheral accessory traits (Carabelli) and central accessory traits (upper cusp 5) may form under different conditions but are dependent on the spacing of principal molar cusps, as summarized in Fig 1.

Comment 14: Lines 436-456: To test hypothesis 6, it might be beneficial to run analyses separately for Carabelli’s trait and cusp 5, rather than lumping them together as 5+ cusped molars. This could be presented as an additional table to Table 6 or even as a supplementary table. As the authors point out, Carabelli’s trait is a peripheral accessory cusp, while cusp 5 is a central accessory cusp. Due to differential placement on the crown, some of the intercusp distance expectations would be different for these two cusps. For Carabelli’s all intercusp distances should be smaller to support the potential of accessory cusp development on the periphery (especially that between cusp 1 and 2). On the other hand, to ensure that a centrally placed accessory cusp 5 can develop, we would expect increased distance between cusps 3 and 4 to ensure the fields of inhibition surrounding these cusps allow additional cusp development. This contradicts predictions for Carabelli’s trait. Therefore, I think it would be helpful to show those analyses separately to make sure that considering the two traits at the same time isn’t obscuring the results obtained.

Response: We thank Reviewer 1 for this suggestion. As recommended, we now test principal cusp configurations derived from PCM predictions separately for the Carabelli and cusp 5 traits. We added wording in the background section (Lines 95-117) and added a new Fig 1 to show the possible configurations that we expect to be associated with each trait. We reworded Hypothesis 6 to be consistent with the new t

---

## [Decision Letter · Decision Letter 1]

9 Jan 2024

PONE-D-23-12197R1Nutritional supplementation, tooth crown size, and trait expression in individuals from Tezonteopan, Mexico.PLOS ONE

Dear Dr. Blankenship-Sefczek,

Thank you for submitting your manuscript to PLOS ONE. After careful consideration, we feel that it has merit but does not fully meet PLOS ONE’s publication criteria as it currently stands. Therefore, we invite you to submit a revised version of the manuscript that addresses the points raised during the review process. There are still concerns over the methodological approach and interpretation of the presented work. Please address these concerns in a revision and a point by point response to these criticisms.

We look forward to receiving your revised manuscript.

Kind regards,

JJ Cray Jr., Ph.D.

Academic Editor

PLOS ONE

Reviewers' comments:

Reviewer's Responses to Questions

**Comments to the Author**

1. If the authors have adequately addressed your comments raised in a previous round of review and you feel that this manuscript is now acceptable for publication, you may indicate that here to bypass the “Comments to the Author” section, enter your conflict of interest statement in the “Confidential to Editor” section, and submit your "Accept" recommendation.

Reviewer #1: (No Response)

Reviewer #2: (No Response)

2. Is the manuscript technically sound, and do the data support the conclusions?

Reviewer #1: Yes

Reviewer #2: Partly

3. Has the statistical analysis been performed appropriately and rigorously? 

Reviewer #1: Yes

Reviewer #2: No

4. Have the authors made all data underlying the findings in their manuscript fully available?

Reviewer #1: Yes

Reviewer #2: Yes

5. Is the manuscript presented in an intelligible fashion and written in standard English?

Reviewer #1: Yes

Reviewer #2: Yes

6. Review Comments to the Author

Reviewer #1: The revised manuscript has adequately addressed the concerns raised during the previous peer-review process. With the revisions, I have a few minor suggestions to the authors:

Lines 106-117: This paragraph contains information identical to what is present in the Figure 1 legend. For brevity, I would recommend introducing Figure 1 in the preceding paragraph discussing the same publication by Ortiz et al. 2018 (Lines 91-105), and eliminating this paragraph.

Lines 267-300: The authors here reference Figure 2 depicting the measurements taken and Figure 3 visualizing the morphological traits scored, but at the bottom of the manuscript, the figures referenced are reversed. I wanted to bring this to the authors’ attention to make sure the appropriate images will be linked to the appropriate manuscript sections and figure legends.

General comment about Results section: As authors adopt the p-value scheme from Muff et al., it would be helpful to remind the reader about the p-value translation of terminology like “little or no evidence” or “weak evidence” throughout the Results section to facilitate the interpretability of results presented. For example, Line 323-324 would look as follows: “We found little evidence (0.1≤p) that M1…”. This would be a simple change, but it would really ease the reader’s ability to contextualize the strength of the evidence for the various hypotheses.

Lines 339-346: At the end of this paragraph, it would be helpful to explain the implications of the moderate effects uncovered by the sign test for the current study and Hypothesis #2 specifically.

Hypotheses 3 and 4 in the Result section: The way the ordinal logistic regression results are presented (by thresholds) in Tables 4 and 5 is very confusing. What is meant exactly by showing “strong evidence for elevated trait expression for the transition” between ASUDAS grades? It may be more helpful to calculate odds ratios and discuss how the odds of developing more pronounced morphological trait expression changes as a function of supplementation.

Lines 479-481: “With respect to PCM expectations, findings were unclear in the comparison of supplemented and non-supplemented groups”. What exactly is this sentence referring to? Was there any analysis that compared PCM expectations by supplement groups directly? Or is this statement in reference to Hypotheses #1-4?

Reviewer #2: The authors have made important changes to this draft of the manuscript, chief of which were reorganization changes that shortened the background and a change in p-value interpretation. While the utility of a binary or “sharp” p-value cutoff is certainly something being discussed in the literature, the authors’ chosen approach is not without pitfalls or important considerations. I strongly encourage the authors to read “The role of p-values in judging strength of evidence and realistic replication expectations” (Gibson 2021, Statistics in Biopharmaceutical Research) for a discussion of the importance of effect size on p-values. And then also “P-values don’t measure evidence” (Lavine 2024, Communications in Statistics – Theory and Methods) and “Why P values are not measures of evidence” (Lakens 2022, Trends in Ecology and Evolution).

Furthermore, Muff et al. (2022) clearly articulates that p-values treated as a continuous indicator of evidence must be accompanied by effective sizes and confidence intervals (“A minimal requirement is that we report effect sizes, CIs, and (if applicable) Bayes factors”). If the authors choose to stick with this approach, please provide CIs and effect sizes.

I also do not think that a continuous approach to p-values negates the need to consider the role of multiple testing. If the authors have evidence to the contrary, please cite. I appreciate the conundrum of an underpowered sample and the risk of Type II error, but the authors must weigh false positives (and therefore claiming a relationship that doesn’t exist) against simply stating that they do not currently have sufficient evidence to demonstrate a relationship. Including p-values as high as 0.10 as “weak evidence” would seem especially likely to lead to misleading conclusions.

In all, I would strongly caution the authors that application of Muff’s cutoffs might be an erroneous approach (see the Lakens letter). Please at least reference the debate and consider rewording statements such as “strong evidence”, “moderate evidence”, etc on the basis of p-value scores.

I appreciate the authors summarizing their results in Table 10, however, I am confused at some of the result summaries. For example, for Hypothesis 2, they state “small but consistent differences” yet only ICD6 for absolute and relative cusp distances shows p-values less than 0.05. For Hypothesis 3b, the authors state there is “moderate evidence of positive overall effect of non-supplementation on Carabelli cusp scores”, but only 3 of the 6 p-values are below 0.10 and only 1 of those three is below 0.05. I would suggest the authors review this table and be more parsimonious in their interpretations.

The abstract lacks a concluding sentence to summarize for the reader the overarching take-away from this investigation.

I am very enthusiastic about this dataset and appreciate what the authors have attempted to do in this study. However, I remain concerned that the associations are mostly weak and may not hold up to further scrutiny via replication study. If re-contextualizing this as a story about this specific population is not feasible due to lack of comparative data, then I would urge the authors to consider framing this as an exploratory analysis that is providing preliminary data to drive future studies.

7. PLOS authors have the option to publish the peer review history of their article (what does this mean?). If published, this will include your full peer review and any attached files.

Reviewer #1: No

Reviewer #2: No

---

## [Author Response · Author response to Decision Letter 1]

22 Feb 2024

JJ Cray Jr., Ph.D.

Academic Editor

PLOS One

Dear Dr. Cray and PLOS One Manuscript Editors and Reviewers, 

My colleagues and I are pleased to submit our revised manuscript entitled “Nutritional supplementation, tooth crown size, and trait expression in individuals from Tezonteopan, Mexico.” We thank you, the Academic Editor, and our reviewers for your thoughtful and insightful comments. 

Most of the comments we received on this revision related to details of our statistical analysis. In response, we have added confidence intervals, effect sizes, and odds ratios to our statistical analyses. Sentences were added throughout the manuscript to acknowledge the use of small sample sizes and to suggest that our study may provide a basis for further exploration in future research studies. 

Reviewer 1:

Comment 1: Lines 106-117: This paragraph contains information identical to what is present in the Figure 1 legend. For brevity, I would recommend introducing Figure 1 in the preceding paragraph discussing the same publication by Ortiz et al. 2018 (Lines 91-105) and eliminating this paragraph.

Response: We added the following sentence to the end of the paragraph referenced, and deleted the paragraph mentioned. Line 105-108: To consider this possibility, Fig 1 illustrates configurations of secondary enamel knots (specifically principal cusps) thought to promote the formation of accessory cusps that have been tested in previous studies (Configurations A-C) [17, 22,23] as well as a fourth configuration (Configuration D).

Comment 2: Lines 267-300: The authors here reference Figure 2 depicting the measurements taken and Figure 3 visualizing the morphological traits scored, but at the bottom of the manuscript, the figures referenced are reversed. I wanted to bring this to the authors’ attention to make sure the appropriate images will be linked to the appropriate manuscript sections and figure legends.

Response: Thank you for bringing this out our attention. File names for the figures have been adjusted to match the true figure and caption.

Comment 3: General comment about Results section: As authors adopt the p-value scheme from Muff et al., it would be helpful to remind the reader about the p-value translation of terminology like “little or no evidence” or “weak evidence” throughout the Results section to facilitate the interpretability of results presented. For example, Line 323-324 would look as follows: “We found little evidence (0.1≤p) that M1…”. This would be a simple change, but it would really ease the reader’s ability to contextualize the strength of the evidence for the various hypotheses.

Response: Thank you for this suggestion to support reader clarity. Reminders of the p-value translation, such as (0.05>p>0.01) for moderate and (p≥ 0.1) for weak evidence were included throughout the results section where evidence was referenced.

Comment 4: Lines 339-346: At the end of this paragraph, it would be helpful to explain the implications of the moderate effects uncovered by the sign test for the current study and Hypothesis #2 specifically.

Response: The following sentence was added to the results for Hypothesis #2. Lines 373-375: Findings from the sign test suggest the non-supplemented group has slightly smaller cusp spacing, in all but one (relative ICD4) dimension compared to the supplemented group.

Comment 5: Hypotheses 3 and 4 in the Result section: The way the ordinal logistic regression results are presented (by thresholds) in Tables 4 and 5 is very confusing. What is meant exactly by showing “strong evidence for elevated trait expression for the transition” between ASUDAS grades? It may be more helpful to calculate odds ratios and discuss how the odds of developing more pronounced morphological trait expression changes as a function of supplementation.

Response: Thank you for this suggestion to clarify our findings. Odds Ratios were computed and included into Tables 4 and 5. The following sentences were added to the results of Hypothesis #3 and Hypothesis #4 respectively. Lines 402-405: The Odds Ratio showed the non-supplemented group had three times greater odds of having a Carabelli trait score of 2 vs a score of less than 2 compared to the supplemented group. This was the only transition in which the 95% confidence limits around the odds ratio did not include 1. And lines 455-457: Overall the Odds Ratio showed that the non-supplemented group had four times greater odds of having elevated cusp 5 expression compared to the supplemented group.

Comment 6: Lines 479-481: “With respect to PCM expectations, findings were unclear in the comparison of supplemented and non-supplemented groups”. What exactly is this sentence referring to? Was there any analysis that compared PCM expectations by supplement groups directly? Or is this statement in reference to Hypotheses #1-4?

Response: Thank you for this suggestion. The referenced sentence was altered as follows to clarify the statement about PCM expectations and supplemented/non-supplemented comparisons. Line 559-561: With respect to PCM expectations, findings were unclear in the comparison of supplemented to non-supplemented groups as outlined in hypotheses 1-4.

Reviewer 2:

Comment 1: The authors have made important changes to this draft of the manuscript, chief of which were reorganization changes that shortened the background and a change in p-value interpretation. While the utility of a binary or “sharp” p-value cutoff is certainly something being discussed in the literature, the authors’ chosen approach is not without pitfalls or important considerations. I strongly encourage the authors to read “The role of p-values in judging strength of evidence and realistic replication expectations” (Gibson 2021, Statistics in Biopharmaceutical Research) for a discussion of the importance of effect size on p-values. And then also “P-values don’t measure evidence” (Lavine 2024, Communications in Statistics – Theory and Methods) and “Why P values are not measures of evidence” (Lakens 2022, Trends in Ecology and Evolution).

Response: Thank you for the opportunity to clarify our use of continuous p-values. We added the following sentences under the Statistical subheading of the Materials and Methods section. Lines 308-343: The variables analyzed for this study (crown size, intercusp spacing, and trait expression) are associated with tightly controlled developmental processes constrained by the demands of precise occlusion. Although these variables would be expected to show developmental integration, a significant portion of their expression could result from independent factors, leading to small covariances in their expression, as is frequently observed with other small biological traits [47]. Thus, assessing each trait individually helps to quantify underlying semi-independent processes that contribute to alterations in final tooth form. Hunter and colleagues [22] found generally low-to-moderate levels of association between Carabelli ASU score and relative intercusp distance (Kendall’s tao ≈ -0.3), suggesting small-to-medium effect size of cusp spacing on Carabelli expression in a much larger sample of 187 individuals. Hunter and colleagues [22] also reported an odds ratio of approximately 8:1 from a logistic regression of Carabelli ASU score vs. relative cusp spacing, which suggests a relatively large effect when calculated over a range of 0.1 units of the relative cusp spacing ratio, which is about half of the observed range of 0.2 relative intercusp distance units between a minimum of 0.5 and a maximum of 0.7. In contrast, differences in the cusp spacing ratio observed in this study between supplemented and non-supplemented groups (see relative intercusp spacing values in Tables 2, 8 and 9 below) are much smaller than 0.2 or even 0.1, and instead range from a minimum of 0.005 to a maximum of 0.04, with most differences between groups below 0.02. Calculated over a 0.02 difference in cusp spacing, the odds ratio drops to 1.5:1, which can be considered a weak effect. Thus, we expect generally small effects on Carabelli expression and presumably other accessory features in this study. With small to medium effect sizes, larger P-values that diverge from the small values traditionally associated with “statistical significance”, are also expected. Given the expected effect sizes and P-values, to discern patterns in our results, we adopt the approach outlined by Muff and colleagues [48] where P-values are used to suggest the relative strength of evidence of predicted relationships rather than as binary cut-offs. Based on the Muff et al. [48] schema, P-values greater than 0.1 imply “little or no evidence”, between 0.1 and 0.05 “weak evidence”, between 0.05 and 0.01 “moderate evidence”, between 0.01 and 0.001 “strong evidence”, and less than 0.001 “very strong evidence”. As stated by Muff and colleagues [49], “little to no evidence” does not mean an absence of evidence or that no effect exists. Due to the rarity of this sample, in which comparisons between nutritional stress levels in humans can be made, our aim is to detect possible trends. We recognize that there exists a continuing debate among statisticians regarding the statistical measure of P-values, including the use of P-values as “strength of evidence”, and the application of strict �-levels [see Schervish 1996; Greenland et al., 2016; Greenland, 2018; Gibson 2021; Muff et al 2022a; Lakens 2022; Muff et al 2022b; Lavine 2024]. As suggested by Muff and colleagues [48-49], we present P-values in conjunction with effect size to show patterns in our results and interpret the possible relationships. We recognize that our data set arises from a small sample size and interpret our findings within this framework. 

Comment 2: Furthermore, Muff et al. (2022) clearly articulates that p-values treated as a continuous indicator of evidence must be accompanied by effective sizes and confidence intervals (“A minimal requirement is that we report effect sizes, CIs, and (if applicable) Bayes factors”). If the authors choose to stick with this approach, please provide CIs and effect sizes.

Response: Thank you for this comment. We added confidence intervals and effect sizes to the results section for each Hypothesis. 

Comment 3: I also do not think that a continuous approach to p-values negates the need to consider the role of multiple testing. If the authors have evidence to the contrary, please cite. I appreciate the conundrum of an underpowered sample and the risk of Type II error, but the authors must weigh false positives (and therefore claiming a relationship that doesn’t exist) against simply stating that they do not currently have sufficient evidence to demonstrate a relationship. Including p-values as high as 0.10 as “weak evidence” would seem especially likely to lead to misleading conclusions.

Response: Thank you for your comment and the opportunity to clarify our approach. We did not mean to imply that the use of continuous p-value negates the need to consider multiple testing. What we are suggesting is that with the small sample and low statistical power that we have, further steps-- such as Bonferroni’s correction-- would increase the possibility of a Type II error to impractical levels (Cabin and Mitchell 2000; Nakagawa 2004; Garcia-Perex 2023) and weaken our ability to detect trends. It is only with multiple testing of large samples with higher power that the probability of making at least one Type I error is a strong concern. Among statisticians, there appears to be no consensus regarding when Bonferroni’s correction should be used (Perneger 1998; Cabin and Mitchell 2000; Nakagawa 2004; Garcia-Perez 2023; we have included full citations at the end of this document). We deliberately use continuous p-values to show trends in our data set, rather than definitive language around supporting or not-supporting our findings. We have added sentences throughout the paper to acknowledge both our underpowered sample as well as the uniqueness of our data set. Our goal is to detect trends in our unique data set. Strict cut-off alpha levels are based on data-specific parameters and are not intended to be universally applied (as noted by Schervish 1996; Greenland et al., 2016; Greenland, 2018; Gibson 2021; Muff et al 2022a; Lakens 2022; Muff et al 2022b; Lavine 2024). In adopting a strict cut-off alpha-level for our study, we would lose the ability to detect nuanced differences associated with the small effect sizes we expect for the biological relationships we are exploring. As Nakagawa (2004:1045) states, the use of effect sizes to emphasize biological significance over strict statistical significance is a more meaningful approach. Not reporting on low power studies contributes to publication bias that could result in a loss of information and hinder the progress of science in any field (as discussed in Nakagawa 2004). 

Comment 4: In all, I would strongly caution the authors that application of Muff’s cutoffs might be an erroneous approach (see the Lakens letter). Please at least reference the debate and consider rewording statements such as “strong evidence”, “moderate evidence”, etc. on the basis of p-value scores.

Response: We added the following sentence to the Statistical Subheading of the Materials and Methods section. Lines 338-343: We recognize that there exists a continuing debate among statisticians regarding the statistical measure of P-values, including the use of P-values as “strength of evidence”, and the application of strict �-levels [see Schervish 1996; Greenland et al., 2016; Greenland, 2018; Gibson 2021; Muff et al 2022a; Laken 2022; Muff et al 2022b; Lavine 2024]. As suggested by Muff and colleagues [2022], we present P-values in conjunction with effect size to show patterns in our results and interpret the possible relationships. We recognize that our data set arises from a small sample size and interpret our findings within this framework.

Comment 5: I appreciate the authors summarizing their results in Table 10, however, I am confused at some of the result summaries. For example, for Hypothesis 2, they state “small but consistent differences” yet only ICD6 for absolute and relative cusp distances shows p-values less than 0.05. For Hypothesis 3b, the authors state there is “moderate evidence of positive overall effect of non-supplementation on Carabelli cusp scores”, but only 3 of the 6 p-values are below 0.10 and only 1 of those three is below 0.05. I would suggest the authors review this table and be more parsimonious in their interpretations.

Response: Thank you for commenting on the wording used in Table 10. In the presentation of Strength of Evidence for Hypothesis 2, we are using “difference” not in terms of statistical significance, but in terms of a difference in value (see Table 2). With the sign test (Table 2), we note a consistency of direction in the differences between intercusp distances. Although these effects are small, they are in the same direction. The alternative hypothesis is that the direction of difference is random, where we would find some positive and some negative values with no overall pattern/direction of difference. That is not the case here, where we show consistent difference in the same direction (all but one value is positive; Table 2). We have added a note at the bottom of Table 10 for clarification. Line 568-569: *Here we refer to differences in value between supplemented and non-supplemented groups of absolute and relative intercusp distances (see Table 2). For hypothesis 3b, we use the phrase “moderate evidence” to be consistent with our use of p-values as a range instead of strict cut-offs.

Comment 6: The abstract lacks a concluding sentence to summarize for the reader the overarching take-away from this investigation.

Response: Thank you for bringing this to our attention. We added the following sentence to the end of the abstract. Lines 43-44: Overall, our findings suggest that nutritional stress may affect accessory cusp expression. 

Comment 7: I am very enthusiastic about this dataset and appreciate what the authors have attempted to do in this study. However, I remain concerned that the associations are mostly weak and may not hold up to further

---

## [Decision Letter · Decision Letter 2]

26 Mar 2024

PONE-D-23-12197R2Nutritional supplementation, tooth crown size, and trait expression in individuals from Tezonteopan, Mexico.PLOS ONE

Dear Dr. Blankenship-Sefczek,

Thank you for submitting your manuscript to PLOS ONE. After careful consideration, we feel that it has merit but does not fully meet PLOS ONE’s publication criteria as it currently stands. Therefore, we invite you to submit a revised version of the manuscript that addresses the points raised during the review process.

There are just a few minor issues that need to be addressed in a revision.

We look forward to receiving your revised manuscript.

Kind regards,

JJ Cray Jr., Ph.D.

Academic Editor

PLOS ONE

Journal Requirements:

Reviewers' comments:

Reviewer's Responses to Questions

**Comments to the Author**

1. If the authors have adequately addressed your comments raised in a previous round of review and you feel that this manuscript is now acceptable for publication, you may indicate that here to bypass the “Comments to the Author” section, enter your conflict of interest statement in the “Confidential to Editor” section, and submit your "Accept" recommendation.

Reviewer #1: (No Response)

Reviewer #2: All comments have been addressed

2. Is the manuscript technically sound, and do the data support the conclusions?

Reviewer #1: Yes

Reviewer #2: Yes

3. Has the statistical analysis been performed appropriately and rigorously? 

Reviewer #1: Yes

Reviewer #2: Yes

4. Have the authors made all data underlying the findings in their manuscript fully available?

Reviewer #1: No

Reviewer #2: Yes

5. Is the manuscript presented in an intelligible fashion and written in standard English?

Reviewer #1: Yes

Reviewer #2: Yes

6. Review Comments to the Author

Reviewer #1: Lines 42-43: This sentence is confusing. What is meant by “strong evidence of mesiodistal closer approximation”? Please consider rephrasing.

Lines 308-343: I recognize this paragraph was prepared in response to Reviewer 2’s comments, but it is incredibly hard to follow. The authors make the argument that the variables examined in this study are “expected to show developmental integration”, but some of the variation can “result from independent factors”. Isn’t the PCM, which authors rely on this paper, a framework which integrates all of these variables to explain final tooth form? I really struggle to follow the first half of this paragraph discussing effect sizes and intercusp ratios. In my opinion, the second half of the paragraphs is the more relevant discuss that directly addresses Reviewer 2’s comments. I would recommend either revising or eliminating the first half of this paragraph and concentrating on the p-value debate here.

Table 3 and Table 7: Since Tables 3 and 7 only provide statistics, it’s hard to understand how morphological trait expression changes between groups. It would be helpful to add mean ASUDAS scores for each group (supplemented/non-supplemented) and tooth (I1/I2/C/M1/M2), similar to how mean intercusp distances and crown sizes are provided in the other tables. This will help the reader see how mean trait expression changes as a function of supplementation in each tooth.

Lines 603-605: It may be worth discussing the crown size differences as it relates to morphological trait expression here (from Tables 8 and 9).

General comment: The authors indicate that the data utilized in this study are available without restriction, but the raw data does not appear to be attached to the supplement and if it’s housed at an external repository, this isn’t indicated or linked. To comply with PLoS One’s data policy, authors should address this.

Reviewer #2: My thanks again to the authors for this thorough revision. The inclusion of odds ratios, CIs and effect sizes has helped greatly in allowing the reader to evaluate the strength of evidence. I now believe this manuscript is ready for publication.

7. PLOS authors have the option to publish the peer review history of their article (what does this mean?). If published, this will include your full peer review and any attached files.

Reviewer #1: No

Reviewer #2: No

---

## [Author Response · Author response to Decision Letter 2]

11 Apr 2024

Journal Requirements: 

Comment 1: Please review your reference list to ensure that it is complete and correct. If you have cited papers that have been retracted, please include the rationale for doing so in the manuscript text, or remove these references and replace them with relevant current references. Any changes to the reference list should be mentioned in the rebuttal letter that accompanies your revised manuscript. If you need to cite a retracted article, indicate the article’s retracted status in the References list and also include a citation and full reference for the retraction notice.

Response: Thank you for your comment regarding citations. There was a duplicate citation (Heaton and Pickering) listed as number 63. This citation has been removed and all subsequent citations have been updated in the discussion (lines 657, 659, and 660) and conclusion (line 706) sections as well as the references section (lines 886-900). 

Reviewer #1: 

Comment 1: Lines 42-43: This sentence is confusing. What is meant by “strong evidence of mesiodistal closer approximation”? Please consider rephrasing.

Response: Lines 42-43 rephrased to: “With both groups pooled together, there was strong evidence of closer mesiodistal distances among principal cusps in molars with accessory cusps, a finding that is consistent with the PCM.”

Comment 2: Lines 308-343: I recognize this paragraph was prepared in response to Reviewer 2’s comments, but it is incredibly hard to follow. The authors make the argument that the variables examined in this study are “expected to show developmental integration”, but some of the variation can “result from independent factors”. Isn’t the PCM, which authors rely on this paper, a framework which integrates all of these variables to explain final tooth form? I really struggle to follow the first half of this paragraph discussing effect sizes and intercusp ratios. In my opinion, the second half of the paragraphs is the more relevant discuss that directly addresses Reviewer 2’s comments. I would recommend either revising or eliminating the first half of this paragraph and concentrating on the p-value debate here.

Response: The addition referred to by reviewer 1 was a concern addressed in response to comments made by reviewer 2. For clarity, we split this long paragraph into three paragraphs spanning lines 308-344. Wording in the last sentence of the first paragraph was edited for further clarity. Lines 313-314: Thus, statistically analyzing each trait individually helps to quantify underlying semi-independent processes that contribute to alterations in final tooth form.

Comment 3: Table 3 and Table 7: Since Tables 3 and 7 only provide statistics, it’s hard to understand how morphological trait expression changes between groups. It would be helpful to add mean ASUDAS scores for each group (supplemented/non-supplemented) and tooth (I1/I2/C/M1/M2), similar to how mean intercusp distances and crown sizes are provided in the other tables. This will help the reader see how mean trait expression changes as a function of supplementation in each tooth.

Response: Average ASUDAS score is not typically presented since the “n” and % are more meaningful values for comparison (see Scott et al 2018; Irish and Guatelli-Steinberg 2003; Scott and Turner 1997 among others). Both “n” and % for trait scores are presented in the table found in the S2 document. For clarification, we added the following line to tables 3 and 7: statistical results presented here are based on data shown in S2.

Comment 4: Lines 603-605: It may be worth discussing the crown size differences as it relates to morphological trait expression here (from Tables 8 and 9).

Response: Lines 603-605 in the discussion pertain to molar intercusp distance and dietary stress whereas results presented in tables 8 and 9 are in reference to the entire sample (not separated into supplemented and non-supplemented groups). Adding the information from tables 8 and 9 in this location of the discussion might confuse these two comparisons. 

Comment 5: General comment: The authors indicate that the data utilized in this study are available without restriction, but the raw data does not appear to be attached to the supplement and if it’s housed at an external repository, this isn’t indicated or linked. To comply with PLoS One’s data policy, authors should address this.

Response: We include the minimal data set (means, SD, CI, frequencies) in the manuscript tables and supporting tables. Following the PLOS ONE data availability statements, we are consistent with data sharing within the field which is to share data that have been processed for statistical analysis. 

Reviewer #2: 

Comment 1: My thanks again to the authors for this thorough revision. The inclusion of odds ratios, CIs and effect sizes has helped greatly in allowing the reader to evaluate the strength of evidence. I now believe this manuscript is ready for publication.

Response: Thank you for comments. We are happy to hear that the revisions we made adequately addressed your concerns.

---

## [Editor Report · Decision Letter 3]

24 May 2024

Nutritional supplementation, tooth crown size, and trait expression in individuals from Tezonteopan, Mexico.

PONE-D-23-12197R3

Dear Dr. Blankenship-Sefczek,

We’re pleased to inform you that your manuscript has been judged scientifically suitable for publication and will be formally accepted for publication once it meets all outstanding technical requirements.

Kind regards,

JJ Cray Jr., Ph.D.

Academic Editor

PLOS ONE
---

## [Editor Report · Acceptance letter]

28 May 2024

PONE-D-23-12197R3 

PLOS ONE

Dear Dr. Blankenship-Sefczek, 

I'm pleased to inform you that your manuscript has been deemed suitable for publication in PLOS ONE. Congratulations! Your manuscript is now being handed over to our production team.

Kind regards, 

on behalf of

Dr. JJ Cray Jr. 

Academic Editor

PLOS ONE